

# Ozone–NO$_x$–VOC Sensitivity of the Lake Michigan Region Inferred from TROPOMI Observations and Ground-Based Measurements

J. Jerrold M. Acdan[1*], R. Bradley Pierce[1,2], Angela F. Dickens[3], Zachariah Adelman[3], Tsengel Nergui[3]

[1]Department of Atmospheric and Oceanic Sciences, University of Wisconsin-Madison, Madison, WI, 53706, United States
[2]Space Science and Engineering Center, University of Wisconsin-Madison, Madison, WI, 53706, United States
[3]Lake Michigan Air Directors Consortium (LADCO), Hillside, IL, 60162, United States

*Correspondence to*: J. Jerrold M. Acdan (acdan@wisc.edu)

**Abstract.** Surface-level ozone (O$_3$) is a secondary air pollutant that has adverse effects on human health. In the troposphere,
O$_3$ is produced in complex cycles of photochemical reactions involving nitrogen oxides (NO$_x$) and volatile organic compounds
(VOCs). Determining if O$_3$ production will be decreased by lowering NO$_x$ emissions ("NO$_x$-sensitive"), VOC emissions
("VOC-sensitive"), or both ("the transition zone") can be done by using the formaldehyde (HCHO; a VOC species) to nitrogen
dioxide (NO$_2$; a component of NO$_x$) concentration ratio ([HCHO]/[NO$_2$]; "FNR"). Generally, lower FNR values indicate VOC
sensitivity while higher values indicate NO$_x$ sensitivity. In this study, we use FNRs calculated from 2019–2021 TROPOspheric
Monitoring Instrument (TROPOMI) satellite data and 2019 Photochemical Assessment Monitoring Station (PAMS) ground-
based data to investigate the ozone–NO$_x$–VOC sensitivity of the Lake Michigan region, an area that regularly exceeds the
United States Environmental Protection Agency's regulatory standards for O$_3$. We find that TROPOMI FNRs are always
greater than PAMS FNRs, indicating that they must be interpreted with different threshold values to infer O$_3$ chemistry
sensitivities. Further analysis of TROPOMI FNRs reveals that during both typical O$_3$ season days and Chicago, Illinois, O$_3$
exceedance days, the average O$_3$ chemistry sensitivity is: (1) VOC-sensitive in the Chicago metropolitan area (CMA), (2)
transitional in the areas surrounding the CMA and up the western Lake Michigan coastline to Milwaukee, Wisconsin, and (3)
NO$_x$-sensitive in the rest of the domain. However, the magnitude of FNR values change during exceedance days, indicating
that areas that are NO$_x$-sensitive (VOC-sensitive) during typical O$_3$ season days increase in NO$_x$-sensitivity (VOC-sensitivity).
Additionally, the transition zone area decreases by 25% on exceedance days. Comparing weekends to weekdays, O$_3$ chemistry
in the Chicago metropolitan area becomes more NO$_x$-sensitive on weekends due lower NO$_x$ emissions. Finally, analysed 10-
meter wind data shows that the lake breeze circulation, which transports high O$_3$ levels from over Lake Michigan to onshore
coastal areas, is stronger during O$_3$ exceedance days compared to typical O$_3$ season days, and there are no major weekday-
weekend differences in the properties of the 10-meter wind field.



## 1 Introduction

Ground-level ozone ($O_3$) is an air pollutant that is known for its harmful effects on public health. Exposure to elevated $O_3$ levels can cause acute respiratory problems and lead to premature death from respiratory and circulatory system illnesses over longer timescales (Jerrett et al., 2009; Turner et al., 2016; Nuvolone et al., 2018). To prevent such adverse effects, the United States Environmental Protection Agency (U.S. EPA) sets a National Ambient Air Quality Standard (NAAQS) for $O_3$, which is currently 70 parts per billion by volume (ppbv) for the fourth-highest daily maximum 8-hour concentration (MDA8),

averaged across three consecutive years (U.S. EPA, 2022a). The U.S. EPA designates counties in the U.S. where $O_3$ concentrations exceed the $O_3$ NAAQS as $O_3$ nonattainment areas (NAAs), and these areas are required by law to develop State Implementation Plans (SIPs) to address the problem. One such region of the U.S. that continues to be in violation of the $O_3$ NAAQS is the Lake Michigan region, including NAAs in the states of Illinois, Indiana, Michigan, and Wisconsin (U.S. EPA, 2022b).


Developing strategies to decrease surface $O_3$ levels is complicated because $O_3$ is a secondary pollutant not directly emitted into the atmosphere. Many studies have shown that the production rate of tropospheric $O_3$ is a non-linear function of the concentrations of the precursor species nitrogen oxides ($NO_x$) and volatile organic compounds (VOCs) (Haagen-Smit, 1952; Milford et al., 1989; Sillman, 1995). In the Lake Michigan region, $NO_x$ emissions primarily come from anthropogenic fossil

fuel combustion, such as diesel and gasoline vehicle usage, electricity generation, and industrial processes (U.S. EPA, 2021). VOC emissions come from both anthropogenic sources (e.g., paint and solvent application) and natural biogenic emissions from plants and crops (U.S. EPA, 2021). Further details about the chemical cycles involving $NO_x$ and VOCs that produce tropospheric $O_3$ can be found in Jacob (1999, 2000), Thornton et al. (2002), Schroeder et al. (2017), and Vermeuel et al. (2019).

Despite the complexities of surface $O_3$ production, $O_3$ formation can be generally classified into two primary chemistry regimes: VOC-sensitive and $NO_x$-sensitive. In the VOC-sensitive regime, VOC emissions reductions decrease organic radical ($RO_2$) formation, which then leads to less cycling with $NO_x$ and thus less $O_3$ formation (Milford et al., 1989; Jin and Holloway, 2015). Conversely, reductions in $NO_x$ emissions in the VOC-sensitive regime can lead to increased $O_3$ formation due to non-linearities in the $O_3$ chemistry. In the $NO_x$-sensitive regime, $NO_x$ emissions reductions decrease the amount of $NO_2$ available

for photolysis, thus reducing the amount of free oxygen atoms that are available to combine with oxygen molecules ($O_2$) to form $O_3$ (Jin and Holloway, 2015). In a third regime between VOC and $NO_x$ sensitivities ("the transition zone"), $O_3$ production can be reduced by decreasing either or both VOC and $NO_x$ emissions (Jin and Holloway, 2015). Therefore, knowing the $O_3$ chemistry sensitivity within a geographical area is informative for regulatory agencies responsible for developing $O_3$ NAAQS attainment strategies through $NO_x$ and VOC emissions control programs.




One of the ways $O_3$ chemistry sensitivities can be identified is through indicator ratios. Sillman (1995) found that the formaldehyde (HCHO) to reactive nitrogen ($NO_y$) concentration ratio ([HCHO]/[$NO_y$]) is a viable indicator for ozone–$NO_x$–VOC sensitivity. HCHO can be used as a proxy for VOC concentrations because many VOC oxidation reactions in $O_3$ chemistry cycles produce HCHO (Sillman, 1995; Valin et al., 2016). Building upon Sillman's work, Tonnesen and Dennis

(2000) found that the formaldehyde to nitrogen dioxide concentration ratio ([HCHO]/[$NO_2$]; referred to as "FNR" for the rest of this paper) is a more useful indicator of ozone–$NO_x$–VOC sensitivity since HCHO and $NO_2$ have similar lifetimes (on the order of hours) while $NO_y$ has a lifetime on the order of days. Using species with similar lifetimes in the indicator ratio better represents the interactions between the catalytic cycles that result in $O_3$ production (Tonnesen and Dennis, 2000).

The FNR indicator ratio is well-suited for satellite data analyses because HCHO and $NO_2$ column densities are both measurable from space. Satellite-derived FNRs from the Global Ozone Monitoring Experiment (GOME), SCanning Imaging Absorption spectroMeter for Atmospheric CartograpHY (SCIAMACHY), and Ozone Monitoring Instrument (OMI) have been used to infer ozone–$NO_x$–VOC sensitivity for many regions around the world (Martin et al., 2004; Duncan et al., 2010; Jin and Holloway, 2015; Chang et al., 2016; Jin et al., 2017; Jin et al., 2020). Despite being applied in many analyses, the threshold

FNR values used to distinguish between VOC-sensitive and $NO_x$-sensitive $O_3$ chemistry varies by study. Much of the early work in this line of research (e.g., Martin et al., 2004; Duncan et al., 2010) used models to estimate FNR threshold values. However, model results can be biased due to errors in input datasets or imperfect representations of physical processes within the models themselves (Brown-Steiner et al., 2015). A more recent study by Jin et al. (2020; referred to as "J20" for the rest of this work) avoids model biases by connecting satellite data directly to ground monitor data to estimate FNR threshold values

for major cities in the U.S., including Chicago, Illinois, in the Lake Michigan region. **Table 1** displays the J20 threshold values for different $O_3$ chemistry sensitivities.

**Table 1: J20 FNR threshold values indicating different $O_3$ chemistry sensitivities for Chicago, Illinois, U.S.**

| Source | VOC-sensitive | Transition zone | $NO_x$-sensitive |
|---|---|---|---|
| Jin et al. 2020 ("J20") | FNR < 3.2 | 3.2 < FNR < 4.1 | FNR > 4.1 |

In this work, we apply the J20 $O_3$ chemistry sensitivity thresholds to satellite FNRs to assess the average 2019-2021 spatial distribution of ozone–$NO_x$–VOC sensitivity in the Lake Michigan region. We calculate FNR values using retrievals from the TROPOspheric Monitoring Instrument (TROPOMI) onboard the Sentinel-5 Precursor (S5P) satellite as well as ground-based measurements from a Photochemical Assessment Monitoring Station (PAMS). After comparing the satellite-based and ground-based FNR values, we further analyse the satellite FNRs to compare typical $O_3$ season days to $O_3$ exceedance days and

weekdays to weekends. We supplement these investigations with an analysis of 10-meter winds to provide meteorological context to the satellite-based results.



## 2 Data & methodology

### 2.1 Satellite data: S5P TROPOMI

The S5P satellite was launched in October 2017 and has a polar, sun-synchronous orbit, providing global daily data
approximately at 13:30 local solar time (Ludewig et al., 2020). TROPOMI is an ultraviolet-visible-near infrared-shortwave
infrared nadir-viewing grating spectrometer onboard the S5P satellite that measures trace gases in the atmosphere as well as
cloud and aerosol properties (Veefkind et al., 2012; van Geffen et al., 2020). The original spatial resolution of TROPOMI data
was 3.5 km x 7 km. Since August 2019, the spatial resolution has been upgraded to 3.5 km x 5.5 km (van Geffen et al., 2020).
TROPOMI retrievals have signal-to-noise ratios (SNRs) that are similar to the Ozone Monitoring Instrument (OMI) but with
much higher spatial resolution (Veefkind et al., 2012; De Smedt et al., 2018). To calculate satellite FNRs, we downloaded S5P
TROPOMI version 1 and 2 tropospheric $NO_2$ (Koninklijk Nederlands Meteorologisch Instituut [KNMI], 2018, 2019, 2021)
and tropospheric HCHO (German Aerospace Center [DLR], 2019, 2020) orbital level 2 (L2) data for the $O_3$ seasons of 2019,
2020, and 2021 from the NASA Goddard Earth Sciences Data and Information Services Center (GES DISC). We define May
to September as the $O_3$ season because this is the period in which most $O_3$ exceedance days occur in Chicago, Illinois, which
is the area that experiences the most exceedances in the Lake Michigan region (**Appendix Table A1**). An $O_3$ exceedance day
is defined as a day in which at least one ground monitor measures MDA8 $O_3$ levels above the U.S. EPA $O_3$ NAAQS (70 ppbv).

Details regarding the TROPOMI $NO_2$ tropospheric vertical column retrieval algorithm are found in van Geffen et al. (2021).
The total uncertainty in $NO_2$ tropospheric vertical column density retrievals is estimated to be between 15–50 % for larger
columns over continental areas (Boersma et al., 2018; van Geffen et al., 2021). Version 1 of TROPOMI $NO_2$ tropospheric
column data had an average bias of -32 % while version 2 had an average bias of -23 % (i.e., are 32 % and 23 % too low,
respectively) when compared to ground-based tropospheric column measurements (van Geffen et al., 2022). Specifics
regarding the TROPOMI HCHO tropospheric vertical column retrieval algorithm are found in De Smedt et al. (2018). The
total uncertainty in HCHO tropospheric vertical column density retrievals is currently estimated to be between 30–60 % in
polluted conditions, including random and systematic slant column contributions (De Smedt et al., 2021). TROPOMI HCHO
data are systematically biased by approximately -25 % (i.e., are 25 % too low) for HCHO columns larger than 8 x $10^{15}$
molecules per square centimeter (mol cm$^{-2}$) when compared to ground-based column measurements (De Smedt et al., 2021).
If the uncertainties in TROPOMI HCHO and $NO_2$ retrievals are uncorrelated, the combined uncertainty in the [HCHO]/[$NO_2$]
ratio is between 34–78 % (see **Appendix B** for calculation). However, Martin et al. (2004) and Duncan et al. (2010) note that
uncertainties from air mass factor calculations (due to clouds, albedo, background corrections, etc.) affect both HCHO and
$NO_2$ retrievals, so these types of errors are likely correlated and tend to cancel out in the [HCHO]/[$NO_2$] ratio. Thus, 34–78 %
should be taken as an upper limit for the uncertainty of FNRs presented in this work.



### 2.1.1 Satellite data processing: S5P TROPOMI composites

TROPOMI retrievals of HCHO are primarily derived from the instrument's spectral Band 3 (range: 310–405 nm), while $NO_2$
retrievals are derived from spectral Band 4 (range: 405–500 nm) (Veefkind et al., 2012; De Smedt et al., 2018; van Geffen et al., 2021). Both bands have a minimum signal-to-noise ratio (SNR) between 800 and 1000 (Veefkind et al., 2012). However, HCHO has an optical density that is an order of magnitude smaller than that of $NO_2$, resulting in HCHO retrievals with lower SNRs than $NO_2$ retrievals (De Smedt et al., 2018). Despite lower SNRs in individual retrievals, Vigouroux et al. (2020) found that monthly mean TROPOMI HCHO composites correlate well with ground-based Fourier-transform infrared (FTIR) station
measurements of HCHO vertical column densities, verifying that the precision of TROPOMI measurements allow the seasonal variability in HCHO levels to be well captured. To reduce the impacts of lower SNRs of TROPOMI HCHO retrievals and to allow for the analysis of general spatial patterns in FNR values, we composited TROPOMI HCHO and $NO_2$ data on monthly and seasonal timescales. We constructed gridded mean TROPOMI HCHO and $NO_2$ 'clear sky' (cloud radiance fraction < 0.5) composites using quality controlled L2 retrievals based on the recommended qa_value > 0.75 for HCHO (De Smedt et al.,
2022) and $NO_2$ (Eskes and Eichmann, 2021). Additional detection limit filters of 1.8 x $10^{15}$ mol $cm^{-2}$ for HCHO (Chance, 2002) and 1.5 x $10^{15}$ mol $cm^{-2}$ for $NO_2$ (Duncan et al., 2010) were applied to the L2 orbital retrievals used in the composites. It is important to note that these detection limit values are for OMI data, but it is assumed that the same values can be applied to TROPOMI data because both instruments have similar signal-to-noise ratios.






The grid onto which the TROPOMI data were composited has a spatial resolution of 12 km x 12 km covering the continental

United States with a Lambert Conformal Conic projection. We spatially subset the composites to the Lake Michigan region (latitude bounds: 41 °N to 45.3 °N; longitude bounds: 88.6 °W to 85.49 °W) for analysis (**Fig. 1**).

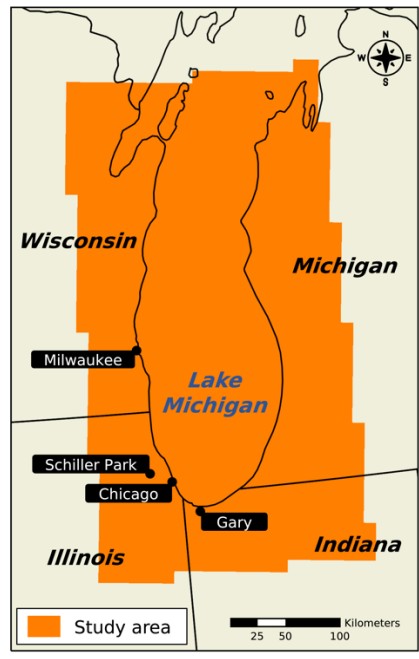

**Figure 1: Map of the Lake Michigan region with the study area highlighted in orange. The locations of cities mentioned in this work**
**are denoted with a black dot.**

We produced TROPOMI HCHO and $NO_2$ composites for four distinct categories: (1) typical $O_3$ season days, (2) $O_3$ exceedance days, (3) weekdays, and (4) weekends (**Table 2**). To create a "typical $O_3$ season day composite", we first created monthly average composites for May to September for the years 2019, 2020, and 2021. We then combined the resulting 15 individual

composites by taking the weighted averages of the monthly composites (based on the number of observations per grid cell) to get an overall 2019–2021 typical $O_3$ season day composite. Next, we next created an "$O_3$ exceedance day composite" by compositing satellite data only on days in which at least one ground monitor in the U.S. EPA Air Quality System (AQS) within the Chicago, Illinois, metropolitan area measured an MDA8 $O_3$ value greater than the NAAQS 70 ppbv standard. For this composite, we deemed Chicago, Illinois, as representative of the Lake Michigan domain since it is the area that experiences

the most $O_3$ exceedances in the region. Therefore, the exceedance day composite included 53 days of TROPOMI data (**Appendix Table A1**). We also created weekday and weekend composites following a similar procedure as that for the $O_3$ season by first compositing weekday and weekend data for individual years and then taking the weighted average to get overall 2019–2021 weekday and weekend composites. For each composite category, we constructed FNR composites by dividing the



composite weighted average HCHO and $NO_2$ values for each grid cell. The resulting FNR composites were then analysed
using the J20 threshold values to infer $O_3$ chemistry sensitivities.

**Table 2: Categories of TROPOMI data composites created in this study.**

| Composite category | Amount of data composited |
| --- | --- |
| Ozone season (May to September) | 459 days (15 months) |
| Chicago $O_3$ exceedance days | 53 days |
| Weekdays | 327 days |
| Weekends | 132 days |

To provide meteorological context to the FNR analyses, we also composited 10-meter height east-west (U) and north-south
(V) winds that are available at each TROPOMI pixel within the downloaded TROPOMI data files. These wind data originally
come from the European Centre for Medium-Range Weather Forecasts (ECMWF) analysis (Eskes et al., 2021). In addition to
wind speed and direction, we also calculated and analysed the divergence of the composited wind field (units: $s^{-1}$) via fourth
order approximations of the horizontal derivatives of U and V (**Eq. 1**–**3**; adapted from Kalnay-Rivas and Hoitsma, 1979):

$$\nabla \cdot \vec{V} = \frac{\partial U}{\partial x} + \frac{\partial V}{\partial y} \tag{1}$$

$$\frac{\partial U}{\partial x} = \frac{4}{3}\left(\frac{U(x+\Delta x)-U(x-\Delta x)}{2\Delta x}\right) - \frac{1}{3}\left(\frac{U(x+2\Delta x)-U(x-2\Delta x)}{4\Delta x}\right) \tag{2}$$

$$\frac{\partial V}{\partial y} = \frac{4}{3}\left(\frac{V(y+\Delta y)-V(y-\Delta y)}{2\Delta y}\right) - \frac{1}{3}\left(\frac{V(y+2\Delta y)-V(y-2\Delta y)}{4\Delta y}\right) \tag{3}$$


**2.2 Ground data & processing: PAMS monitoring site**

To provide further context to TROPOMI-based FNRs, we also calculated FNRs from ground-based data. We obtained
Photochemical Assessment Monitoring Station (PAMS) data for the year 2019, which contains surface HCHO and $NO_2$
measurements for various sites across the U.S. To compare surface FNRs to the satellite FNRs, we considered only sites within
the Lake Michigan domain that reported concurrent HCHO and $NO_2$ measurements during the $O_3$ season. This resulted in a
single PAMS location for analysis: Site 3103 in Schiller Park, Cook County, Illinois (41.965 ºN, 87.876 ºW). The location of
this PAMS site is adjacent to the Chicago O'Hare International airport and major highways.



At this site, HCHO measurements were taken every 6 days in 24-hour sampling intervals while $NO_2$ measurements were taken
every day in 1-hour intervals. To process the data, we first converted the HCHO data from their original units of micrograms
per cubic meter ($\mu g/m^3$) to the same units as the $NO_2$ data (ppb) assuming standard pressure and temperature (because measured
pressure and temperature were not available in the downloaded PAMS dataset). Next, we filtered the $NO_2$ data to include only
the days with concurrent HCHO measurements. These values were then averaged to get individual monthly HCHO and $NO_2$
values. The monthly values were then divided to get monthly FNR values to compare them to the previously generated monthly
composites of TROPOMI data. All available data were used to calculate monthly means for both TROPOMI and PAMS FNR
values (i.e., the TROPOMI FNR composites used in the comparison were not subsampled to include only days in which PAMS
data were available).









# 3 Results & discussion

## 3.1 Comparison of TROPOMI satellite-based and PAMS ground-based FNRs

**Figure 2a** is a time series plot of HCHO and $NO_2$ ground measurements at PAMS Site 3103 in Schiller Park, Illinois, during the 2019 $O_3$ season. The orange dots represent the 24-hour samples of HCHO, sampled every sixth day, while the smaller blue dots indicate the hourly $NO_2$ measurements on those days. The larger blue dots are the daily average of the hourly $NO_2$ measurements, which show that a significant amount of variability is lost when averaging over 24-hour timescales. Since both $NO_2$ and HCHO photolysis follow similar diurnal cycles (Ruiz-Suarez et al., 1993), 24-hour sampling also obscures the

fluctuations in atmospheric HCHO levels that occur on shorter timescales. **Figure 2b** displays the monthly mean of the data plotted in **Figure 2a**. Monthly mean $NO_2$ levels are greater than HCHO levels for all months during the 2019 $O_3$ season. **Figure 2c** shows a time series comparison between the PAMS ground-based and TROPOMI satellite-derived composite FNR values at Schiller Park, Illinois. All ground-based monthly FNR values are less than 1, ranging from a minimum of 0.25 (May) to a maximum of 0.68 (July). For all months, the TROPOMI FNR values (ranging from a minimum of 1.76 [May] to a maximum

of 2.87 [August]) are greater than the PAMS FNR values.

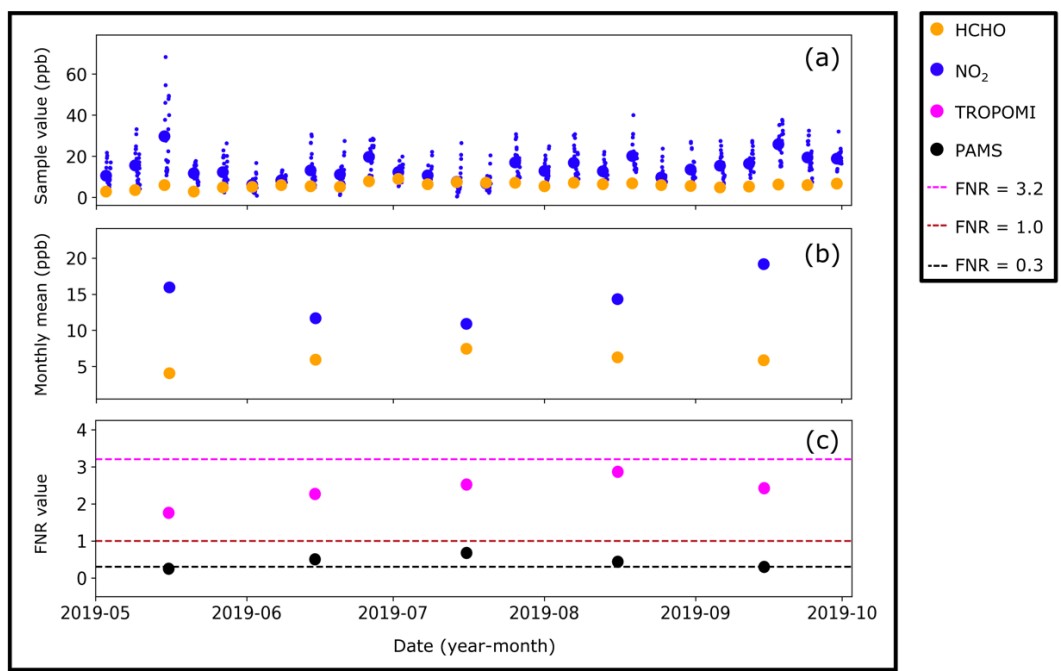

**Figure 2: (a) HCHO (orange) and $NO_2$ (blue) PAMS surface measurements, (b) monthly averaged HCHO (orange) and $NO_2$ (blue) PAMS values, and (c) FNRs calculated from PAMS measurements (black) and TROPOMI retrievals (pink). Areas beneath the**
**dashed pink and black lines represent FNRs indicating VOC-sensitive $O_3$ chemistry based on the Jin et al. (2020) and Blanchard (2020) studies, respectively. The area between the dashed brown and black lines represents FNRs indicating transitional $O_3$ chemistry based on Blanchard (2020).**





The discrepancy between the PAMS and TROPOMI FNR values could be due to many factors including, but not limited to:
(1) the relatively long PAMS HCHO and $NO_2$ sampling duration compared to the instantaneous TROPOMI sampling at 13:30 local time (near the time of peak FNR values; Blanchard, 2020), (2) vertical profile differences between HCHO and $NO_2$ that make surface values that PAMS measures different than tropospheric column values that TROPOMI measures (Schroeder et al., 2017), and (3) high biases in monitored $NO_2$ due to the use of a nonselective detector (Blanchard, 2020). Regardless, this analysis shows that different sets of $O_3$ chemistry sensitivity threshold values must be used to interpret ground-based and satellite-based FNRs. A study by Blanchard (2020) argues that the FNR value indicating the transition between VOC-sensitive and $NO_x$-sensitive $O_3$ chemistry lies between ~0.3–1.0 for the Lake Michigan region. The argument is based on: (1) the transitional $O_3$ chemistry FNR thresholds found in previous studies (e.g., 0.8–1.8 in Tonnesen and Dennis [2000]; 1–2 in Duncan et al. [2010]; 0.7–2.3 in Schroeder et al. [2017]; and 3.2–4.1 in Jin et al. [2020]), (2) the reasons listed above for discrepancies between PAMS and TROPOMI FNRs, and (3) original analyses of surface observations in the Lake Michigan region. If we apply the Blanchard (2020) thresholds to the surface FNRs calculated using PAMS data, we see that $O_3$ production was VOC-sensitive in May and September and transitional in June, July, and August in 2019 at this PAMS site.

The J20 thresholds were derived by connecting satellite-derived FNRs to high $O_3$ event probabilities calculated from ground monitor data. Therefore, the J20 thresholds are well suited to interpret TROPOMI-derived FNRs. If we apply the J20 thresholds to the TROPOMI data, we see that $O_3$ production at this PAMS site is VOC-sensitive for the entire study period in 2019. Comparisons between the ground-based and satellite-based FNRs indicate different $O_3$ chemistry sensitivities from June to August (transitional versus VOC-sensitive, respectively). However, there is some uncertainty with the true lower end value of the Blanchard (2020) transitional $O_3$ chemistry FNR threshold range (~0.3–1.0). If the true lower end value were greater, such as 0.7 reported by Schroeder et al. (2017), then the PAMS FNRs would also indicate that $O_3$ production is VOC-sensitive at this site for the entire study period in 2019.





## 3.2 Comparison of O₃ season and O₃ exceedance day TROPOMI composites

**Figure 3** displays composite mean $NO_2$ levels derived from TROPOMI data during the ozone season ('OS'; **Fig. 3a**) and Chicago ozone exceedance days ('Ex'; **Fig. 3b**), along with mean 10-meter wind vectors. During the ozone season, mean $NO_2$ levels range from $1.95 \times 10^{15}$ to $5.69 \times 10^{15}$ mol cm$^{-2}$, with a regional average of $2.44 \times 10^{15}$ mol cm$^{-2}$. During Chicago exceedance days, mean $NO_2$ levels range from $1.84 \times 10^{15}$ to $7.60 \times 10^{15}$ mol cm$^{-2}$, with a regional average of $2.51 \times 10^{15}$ mol cm$^{-2}$. Additional statistics for these composites can be found in **Appendix Table C1**. The spatial distribution of $NO_2$ levels in

both composites shows a clear hotspot of emissions in the Chicago, Illinois, metropolitan area (CMA). **Figure 3c**, the difference between the exceedance day and ozone season composites, shows that mean $NO_2$ levels are higher in the CMA and north along the western Lake Michigan coastline during exceedance days, with the greatest increase of $1.91 \times 10^{15}$ mol cm$^{-2}$ in the urban core of Chicago. Further research is needed to investigate reasons why $NO_2$ levels are higher in the CMA during Chicago O₃ exceedance days.


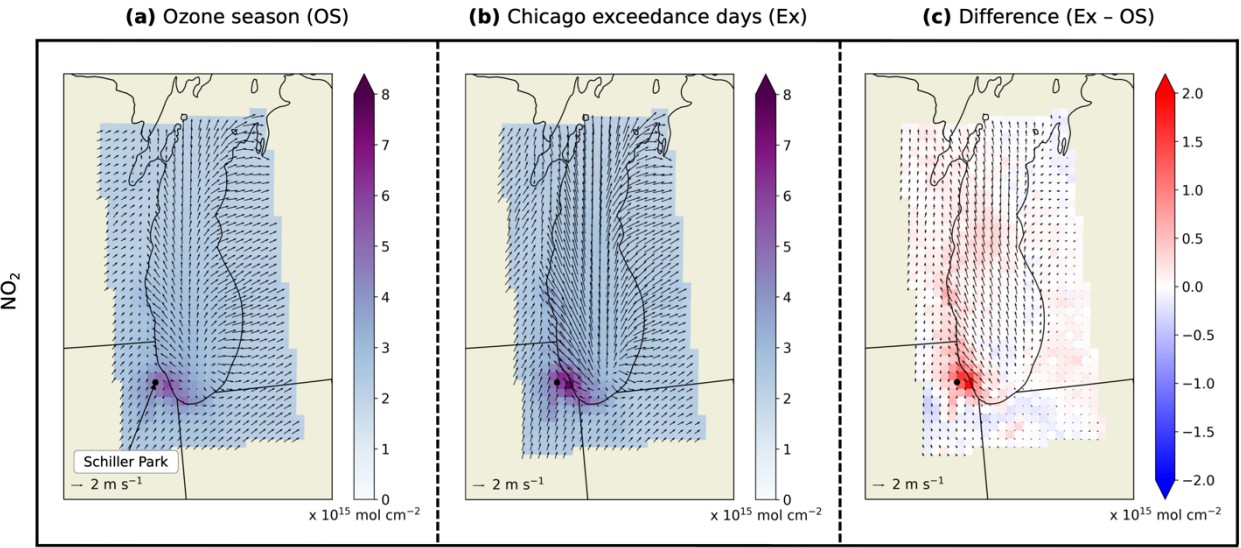

**Figure 3: TROPOMI-derived composites of 2019–2021 mean tropospheric NO₂ in the Lake Michigan region during: (a) the ozone season (OS), (b) Chicago ozone exceedance days (Ex), and (c) the difference between them (Ex – OS). Mean 10-meter winds are represented by arrows.**






**Figure 4** displays composite mean HCHO levels derived from TROPOMI data during the ozone season (**Fig. 4a**) and Chicago exceedance days (**Fig. 4b**), along with mean 10-meter wind vectors. During the ozone season, mean HCHO levels range from $10.2 \times 10^{15}$ to $13.8 \times 10^{15}$ mol cm$^{-2}$, with a regional average of $11.9 \times 10^{15}$ mol cm$^{-2}$. During Chicago exceedance days, mean HCHO levels range from $9.64 \times 10^{15}$ to $16.2 \times 10^{15}$ mol cm$^{-2}$, with a regional average of $13.0 \times 10^{15}$ mol cm$^{-2}$. Additional

statistics for these composites can be found in **Appendix Table C1**. In general, the spatial distribution of HCHO in both composites is relatively homogeneous (compared to $NO_2$), though higher levels can be seen over the southern part of the lake and along the eastern Lake Michigan coastline. **Figure 4c**, the difference between the Chicago exceedance day and ozone season composites, shows that mean HCHO levels are higher during Chicago ozone exceedance days, with an average regional increase of $1.13 \times 10^{15}$ mol cm$^{-2}$. Because positive differences occur over the entire domain (as opposed to occurring in

localized hotspots), the higher HCHO abundances are likely due to increased temperatures during $O_3$ exceedance events, which lead to increased biogenic VOC emissions and thus increased $O_3$ production (Sillman and Samson, 1995).

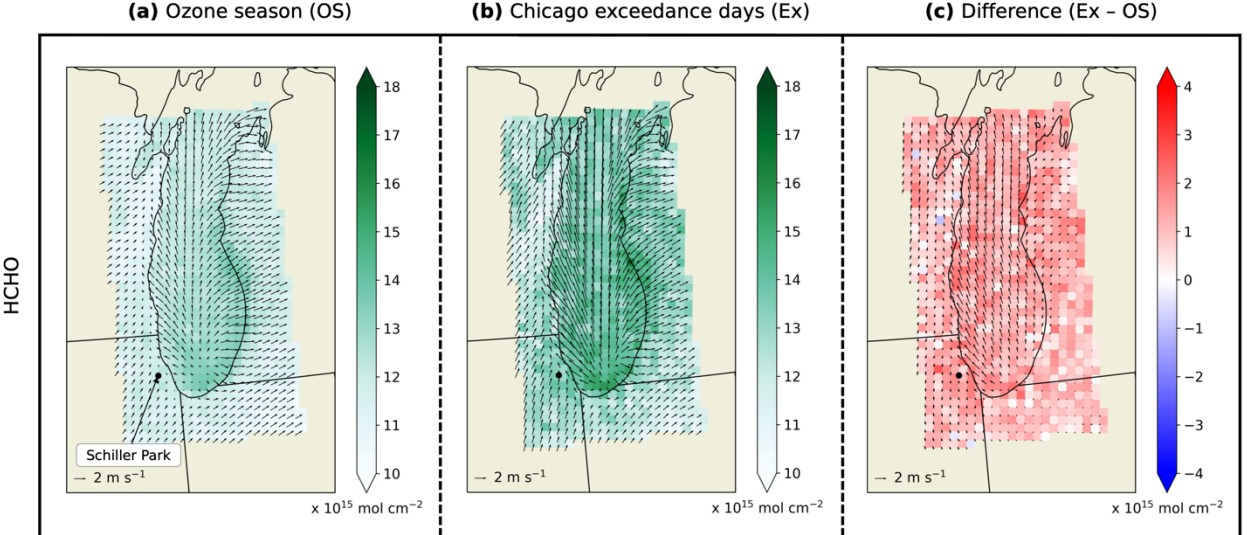

**Figure 4: TROPOMI-derived composites of 2019–2021 mean tropospheric HCHO in the Lake Michigan region during: (a) the ozone**
**season (OS), (b) Chicago ozone exceedance days (Ex), and (c) the difference between them (Ex – OS). Mean 10-meter winds are represented by arrows.**




**Figure 5** displays composite mean FNR values during the ozone season (**Fig. 5a**) and Chicago exceedance days (**Fig. 5b**), along with mean 10-meter wind vectors. During the ozone season, mean FNR values range from 2.12 to 6.12, with a regional average of 4.99. During Chicago exceedance days, mean FNR values range from 1.84 to 7.11, with a regional average of 5.34. Additional statistics for these composites can be found in **Appendix Table C1**. In both composites, the lowest FNR values occur in the CMA, its surroundings, and north along the eastern Illinois and Wisconsin shorelines. The highest values are found along the western Michigan shoreline. **Figure 5c**, the difference between the Chicago exceedance day and ozone season composites, shows that mean FNR values are lower in the urban core of Chicago and in some parts north along the western Lake Michigan coastline during exceedance days, indicating more VOC-sensitive $O_3$ chemistry relative to all days across the season. However, for most of the region, mean FNR values are higher on exceedance days (with a regional average increase of 1.54), indicating increasingly $NO_x$-sensitive $O_3$ chemistry.

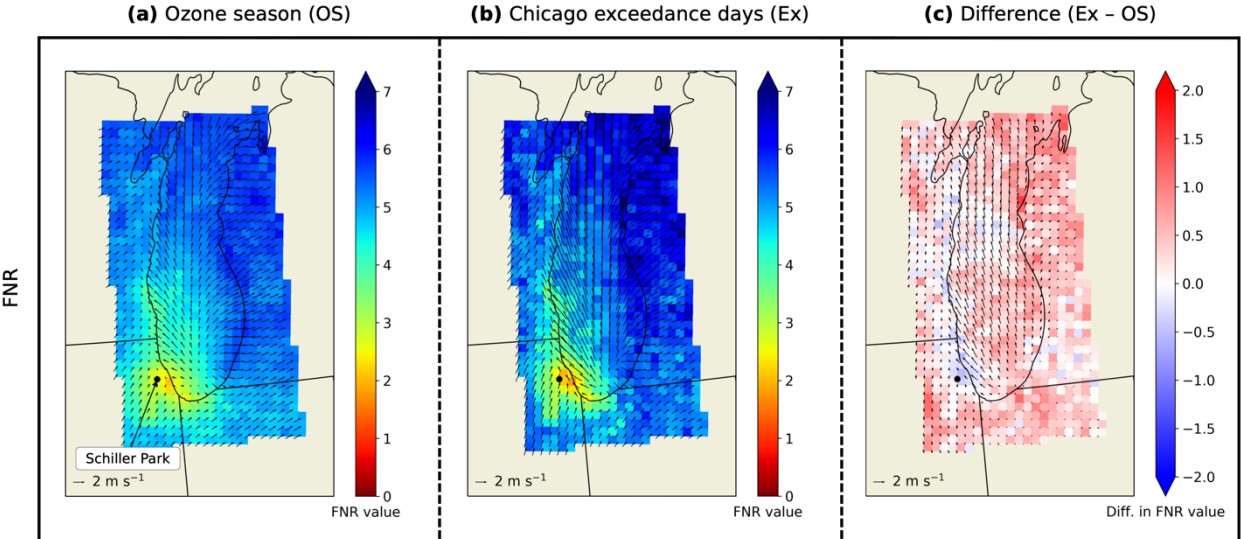

**Figure 5: TROPOMI-derived 2019–2021 FNR values in the Lake Michigan region during: (a) the ozone season (OS), (b) Chicago ozone exceedance days (Ex), and (c) the difference between them (Ex – OS). Mean 10-meter winds are represented by arrows.**



When the mean FNR values are interpreted using the J20 thresholds, we see similar spatial distributions of $O_3$ chemistry
345    sensitivities between typical ozone season days (**Fig. 6a**) and Chicago exceedance days (**Fig. 6b**). For both composites, the
Chicago metropolitan area is VOC-sensitive, while its surroundings and north along the western shoreline of Lake Michigan
up to Milwaukee are in the transition zone between VOC and $NO_x$ sensitivities. The rest of the domain is $NO_x$-sensitive. The
spatial distributions of $O_3$ chemistry sensitivities are similar for two reasons: (1) higher $NO_2$ levels in Chicago during
exceedance days counterbalances higher regional HCHO concentrations, thus making the Chicago metropolitan area remain
350    VOC-sensitive (note that the urban core of Chicago becomes even more VOC-sensitive as indicated by decreases in FNR
values seen in **Fig.5c**), and (2) the largest increases in FNR values (indicating increasing $NO_x$ sensitivity) are outside of
Chicago in the regions that are already $NO_x$-sensitive. It is interesting to note that the area classified as having transitional
chemistry changes from 7.3 % during the ozone season to 5.5 % on Chicago exceedance days, which is a decrease in value of
25 %. This is due to increased HCHO levels that shift transition zone areas in the ozone season composite to $NO_x$-sensitive
areas in the exceedance day composite.

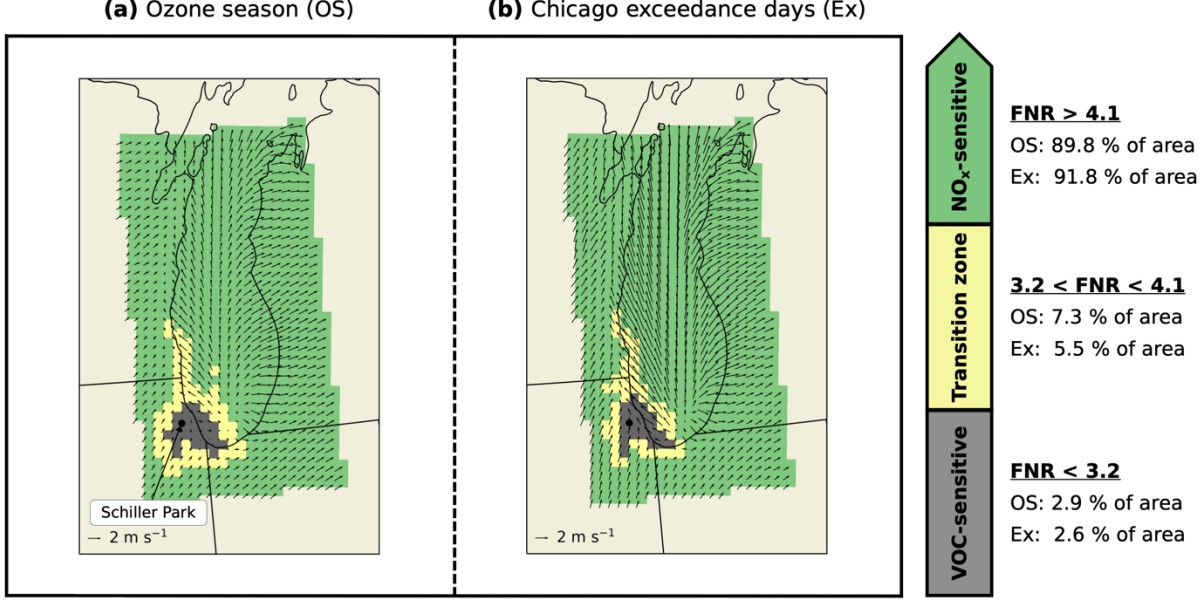

**Figure 6: J20 threshold interpretation of 2019–2021 ozone–$NO_x$–VOC sensitivity in the Lake Michigan region during: (a) the ozone
season and (b) Chicago ozone exceedance days. Mean 10-meter winds are represented by arrows.**


These results are similar to those of Vermeuel et al. (2019) and Acdan et al. (2020), who used chemical box modeling to
investigate the $O_3$ production sensitivity of air parcels as they traveled on $O_3$ exceedance days (June 2/4/11/12/15, 2017) from
their Chicago-Gary (Illinois-Indiana) urban source regions to over Lake Michigan, and then north along the coast to the border
between Illinois and Wisconsin. Their findings show that $O_3$ production within the plumes transitioned from having more



VOC-sensitive $O_3$ chemistry in the Chicago-Gary urban source regions to having more $NO_x$-sensitive $O_3$ chemistry as they advected north along the Lake Michigan coastline. Our results and those of Vermeuel et al. (2019) and Acdan et al. (2020) find a general south-north gradient in $O_3$ chemistry that transitions toward less VOC sensitivity/more $NO_x$ sensitivity starting from the south in the Chicago metropolitan area and going north along the western Lake Michigan shoreline.









**Figure 7** displays plots of mean 10-meter wind vectors and the mean divergence of the wind field during the ozone season
(**Fig. 7a**) and Chicago exceedance days (**Fig. 7b**). In both composites, the average wind pattern along the Lake Michigan
coastline (during the TROPOMI satellite overpass time) resembles a thermally direct circulation known as the lake breeze in
which the wind blows from over the lake and onto the land (Lyons, 1972; Laird et al., 2001). The mean wind field divergence
values occur in two regimes: (1) positive values indicating divergence, which strongly occurs over the lake, and (2) negative
values indicating convergence, which occurs most strongly along the western Lake Michigan coastline during the study period.
In the difference plot (**Fig. 7c**), mean divergence values are more negative along the western Lake Michigan coastline and are
more positive over the southern part of the lake during exceedance days. This indicates that the lake breeze circulation is
stronger during Chicago exceedance days relative to all days across the $O_3$ season. Because the lake breeze is a thermally direct
circulation, the strengthening during Chicago exceedance days suggests that land temperatures are warmer during exceedance
days. This result is consistent with the finding that mean HCHO levels are greater during exceedance days. Additionally, these
findings are also in alignment with other studies that find that $O_3$ is often produced when both $NO_x$ and VOCs are present in a
shallow marine boundary layer above Lake Michigan; the high $O_3$ concentrations are then pushed onshore by the lake breeze,
resulting in counties along the lakeshore having elevated $O_3$ levels (Dye et al., 1995; Stanier et al., 2021; Cleary et al., 2022;
Wagner et al., 2022). Statistics for these composites can be found in **Appendix Table C1**.

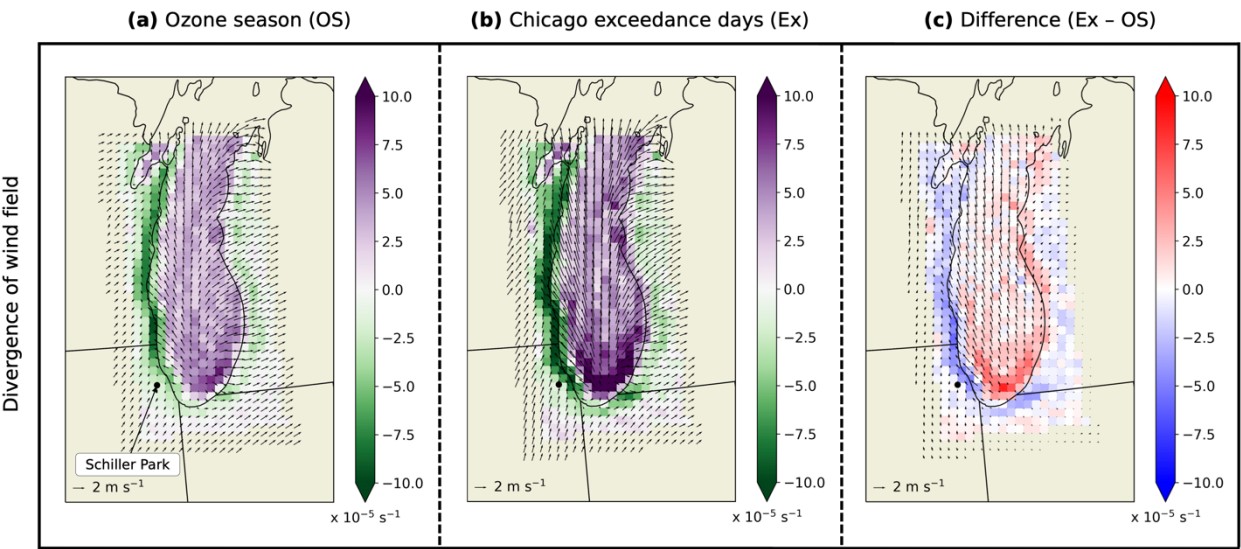


**Figure 7: Divergence values of the wind field (calculated from 2019–2021 mean wind vectors) in the Lake Michigan region during:
(a) the ozone season (OS), (b) Chicago ozone exceedance days (Ex), and (c) the difference between them (Ex – OS). Mean 10-meter
winds are represented by arrows. In (a) and (b), positive (purple) values indicate divergence while negative (green) values indicate
convergence.**






**Figure 8** shows histogram plots of the TROPOMI composite values of mean $NO_2$, HCHO, FNR, and wind divergence values during the ozone season and Chicago exceedance days. Two-sample Kolmogorov-Smirnov tests indicate that the differences between the ozone season and Chicago exceedance day distributions are statistically significant for all four parameters at the

99 % confidence level. These tests and histogram comparison plots provide more detail to the previous analyses. $NO_2$ values (**Fig. 8a**) look similar in both composites. However, there is a notable increase in the maximum $NO_2$ value from $5.69 \times 10^{15}$ to $7.60 \times 10^{15}$ mol $cm^{-2}$ during Chicago exceedance days. The comparison of HCHO values (**Fig. 8b**) shows a large shift in HCHO levels to higher values during Chicago exceedance days. Statistically higher FNR values on Chicago exceedance days (**Fig. 8c**) are driven by higher regional HCHO concentrations (as opposed to changes in $NO_2$ levels). Finally, the widening of the

wind divergence distribution to more negative values and greater positive values (**Fig. 8d**) further demonstrates the strengthening of the lake breeze circulation during exceedance days.

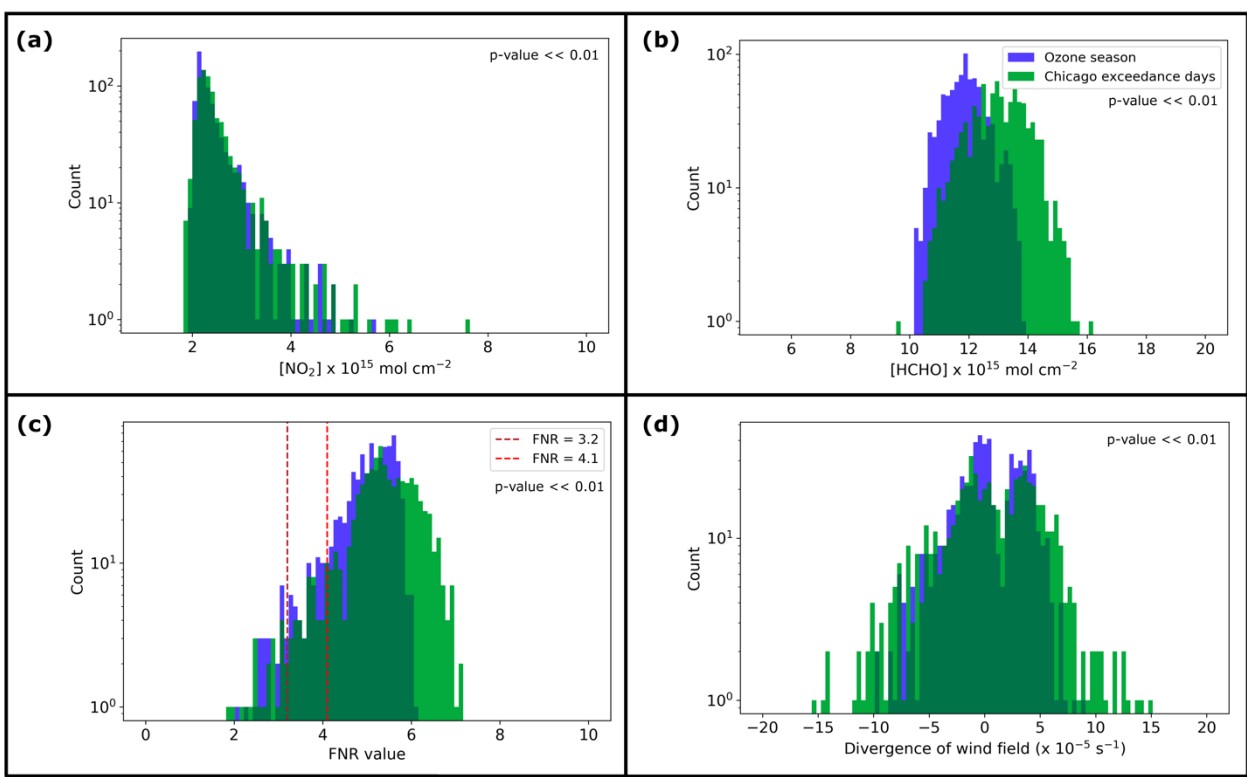

**Figure 8: Histogram plots comparing the distributions of (blue) ozone season values and (green) Chicago ozone exceedance day values (from Figures 3, 4, 5, and 7) of: (a) NO₂, (b) HCHO, (c) FNRs, and (d) wind field divergence.**




### 3.3 Comparison of weekday and weekend TROPOMI composites

**Figure 9** displays composite mean $NO_2$ levels derived from TROPOMI data during weekdays (**Fig. 9a**) and weekends (**Fig. 9b**), along with mean 10-meter wind vectors. During weekdays, mean $NO_2$ levels range from $1.91 \times 10^{15}$ to $6.45 \times 10^{15}$ mol cm$^{-2}$, with a regional average of $2.49 \times 10^{15}$ mol cm$^{-2}$. During weekends, mean $NO_2$ levels range from $1.82 \times 10^{15}$ to $4.11 \times$

$10^{15}$ mol cm$^{-2}$, with a regional average of $2.31 \times 10^{15}$ mol cm$^{-2}$. Additional statistics for these composites can be found in **Appendix Table C2**. **Figure 9c**, the difference between the weekend and weekday composites, shows significantly lower mean $NO_2$ levels in the Chicago metropolitan area on weekends. The regional average change is a decrease of $0.18 \times 10^{15}$ mol cm$^{-2}$ on weekends, with the greatest decrease of $2.60 \times 10^{15}$ mol cm$^{-2}$ occurring in the urban core of Chicago. This result is expected as $NO_x$ emissions generally decrease due to less road traffic volume on weekends, especially from heavy-duty diesel

trucks (Demetillo et al., 2021).

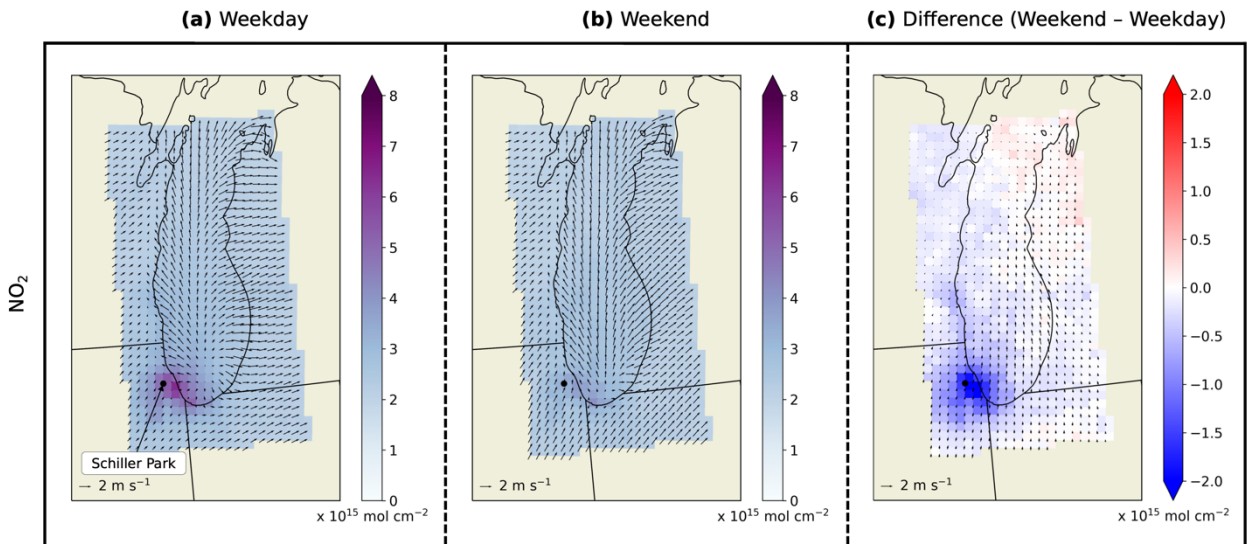

**Figure 9: TROPOMI-derived composites of 2019–2021 mean tropospheric $NO_2$ in the Lake Michigan region during: (a) weekdays, (b) weekends, and (c) the difference between them (weekend – weekday). Mean 10-meter winds are represented by arrows.**





**Figure 10** displays composite mean HCHO levels derived from TROPOMI data during weekdays (**Fig. 10a**) and weekends (**Fig. 10b**), along with mean 10-meter wind vectors. During weekdays, mean HCHO levels range from 10.0 x $10^{15}$ to 13.9 x $10^{15}$ mol cm$^{-2}$, with a regional average of 11.8 x $10^{15}$ mol cm$^{-2}$. During weekends, mean HCHO levels range from 9.74 x $10^{15}$

to 14.3 x $10^{15}$ mol cm$^{-2}$, with a regional average of 12.0 x $10^{15}$ mol cm$^{-2}$. Additional statistics for these composites can be found in **Appendix Table C2**. **Figure 10c**, the difference between the weekend and weekday composites, shows that changes in HCHO levels are mixed as some areas have higher HCHO concentrations on weekends while others have lower concentrations. The average difference is a small increase of 0.21 x $10^{15}$ mol cm$^{-2}$ on weekends. This slight increase could be due to $NO_x$ emissions differences between weekdays and weekends. For example, the principal sink of $NO_x$ is oxidation by hydroxyl

radicals (·OH) to nitric acid ($HNO_3$); therefore, lower $NO_x$ levels could lead to increased ·OH concentrations and thus increased production of secondary HCHO (Jacob, 1999). Further research is needed to investigate this hypothesis.

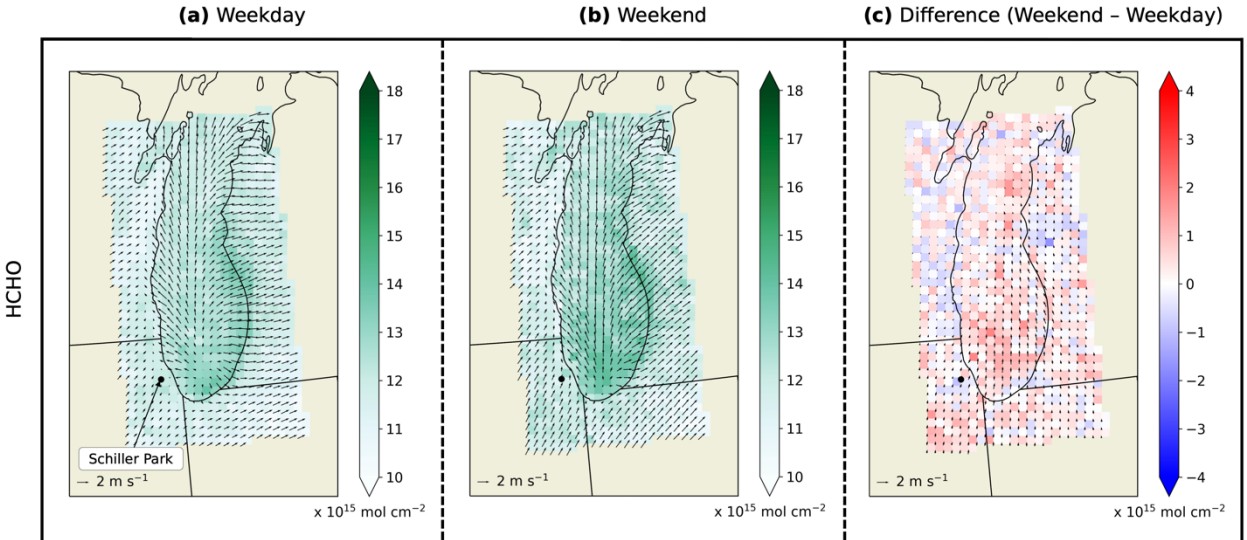

**Figure 10: TROPOMI-derived composites of 2019–2021 mean tropospheric HCHO in the Lake Michigan region during: (a) weekdays, (b) weekends, and (c) the difference between them (weekend – weekday). Mean 10-meter winds are represented by arrows.**





**Figure 11** displays composite mean FNR values during weekdays (**Fig. 11a**) and weekends (**Fig. 11b**), along with mean 10-meter winds. During weekdays, mean FNR values range from 1.88 to 6.23, with a regional average of 4.90. During weekends, mean FNR values range from 2.94 to 6.53, with a regional average of 5.26. Additional statistics for these composites can be found in **Appendix Table C2**. **Figure 11c**, the difference between the weekend and weekday composites, shows that mean FNR values increase for much of the domain on weekends, with the greatest increases in the Chicago metropolitan area, indicating greater $O_3$ chemistry sensitivity to $NO_x$.

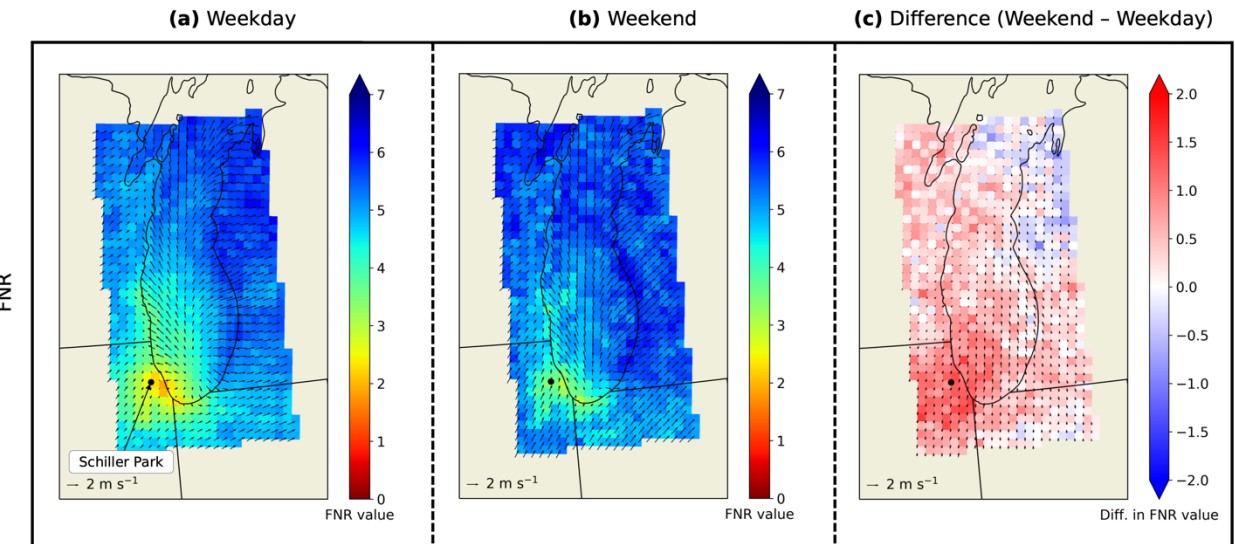

**Figure 11: TROPOMI-derived 2019–2021 FNR values in the Lake Michigan region during: (a) weekdays, (b) weekends, and (c) the difference between them (weekend – weekday). Mean 10-meter winds are represented by arrows.**





When the FNR values are interpreted using the J20 thresholds, we see notable geographical differences in $O_3$ chemistry
sensitivity between weekdays (**Fig. 12a**) and weekends (**Fig. 12b**). The area classified as VOC-sensitive decreases from 4.3
% on weekdays to 0.2 % on weekends; only the urban core of Chicago, Illinois, and industrial areas near Gary, Indiana, remain
VOC-sensitive. The transition zone also decreases from 9.1 % to 3.4 % of the area on weekends, with only grid boxes
containing the CMA, Milwaukee (Wisconsin), and Gary (Indiana) indicating transitional chemistry. The area classified as
$NO_x$-sensitive increases by about 10 % on weekends, largely due to the areas surrounding the Chicago metropolitan area and
the Lake Michigan coastline between Illinois and Wisconsin changing from VOC-sensitive or transitional to becoming $NO_x$-
sensitive.

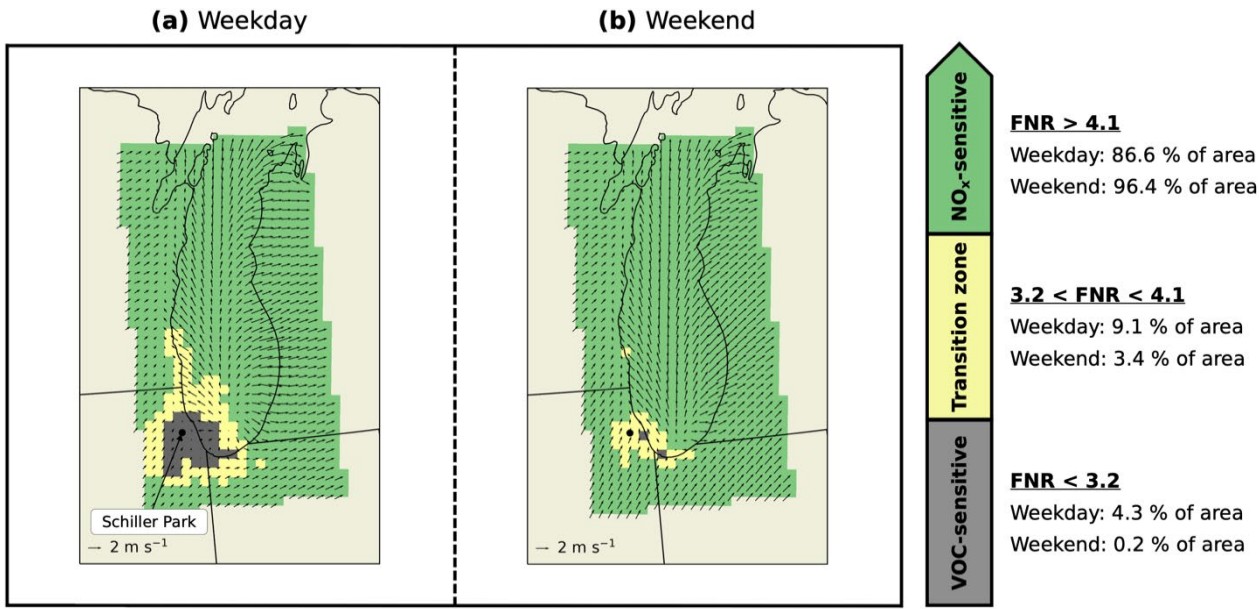

**Figure 12: J20 threshold interpretation of 2019–2021 ozone–$NO_x$–VOC sensitivity in the Lake Michigan region during: (a) weekdays**
**and (b) weekends. Mean 10-meter winds are represented by arrows.**






**Figure 13** displays plots of mean 10-meter wind vectors and the mean divergence of the wind field during weekdays (**Fig. 13a**) and weekends (**Fig. 13b**). Similar to the ozone season and Chicago ozone exceedance day composites, the weekday and weekend composites indicate that the average wind pattern along the Lake Michigan coastline (during the TROPOMI satellite overpass time) resembles a lake breeze circulation. The difference between the composites (**Fig. 13c**) reveals relatively small differences between weekends and weekdays in the mean divergence of the wind field (compared to the difference between the ozone season and Chicago exceedance day composites). Additionally, the average wind speed and direction is approximately the same between weekdays and weekends, as expected. Statistics for these composites can be found in **Appendix Table C2**.

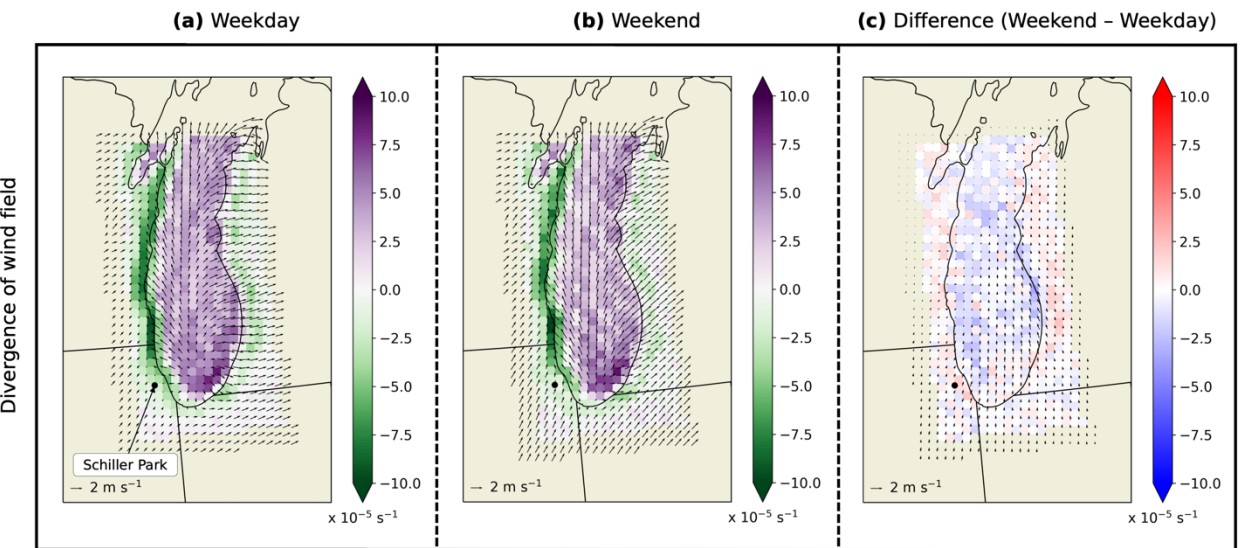

**Figure 13: Divergence values of the wind field (calculated from 2019–2021 mean 10-meter wind vectors) in the Lake Michigan region during: (a) weekdays, (b) weekends, and (c) the difference between them (weekend – weekday). Mean 10-meter winds are represented by arrows. In (a) and (b), positive (purple) values indicate divergence while negative (green) values indicate convergence.**



**Figure 14** shows histogram plots of the TROPOMI composites values during weekdays (blue) and weekends (green). Two-sample Kolmogorov-Smirnov tests indicate that the differences between the weekday and weekend distributions are
statistically significant for mean $NO_2$, HCHO, and FNR values at the 99 % confidence level, while the distributional differences for mean wind divergence values are not statistically significant at the 99 % confidence level. $NO_2$ levels are shifted to lower values on weekends; in particular, the weekend histogram lacks the tail of higher $NO_2$ values as compared to the weekday histogram (**Fig. 14a**). The comparison of HCHO values shows a slight shift to higher HCHO levels during weekends (**Fig. 14b**). On weekends, FNR values are shifted to the right (**Fig. 14c**) due to both slightly higher HCHO and lower $NO_2$ values.
The smaller low tail of the FNR weekend histogram is driven by shifts to higher FNR values in the Chicago metropolitan area and its surroundings primarily due lower $NO_2$ levels. Finally, there does not appear to be a consistent pattern in the differences between wind divergence on weekdays and weekends (**Fig. 14d**).

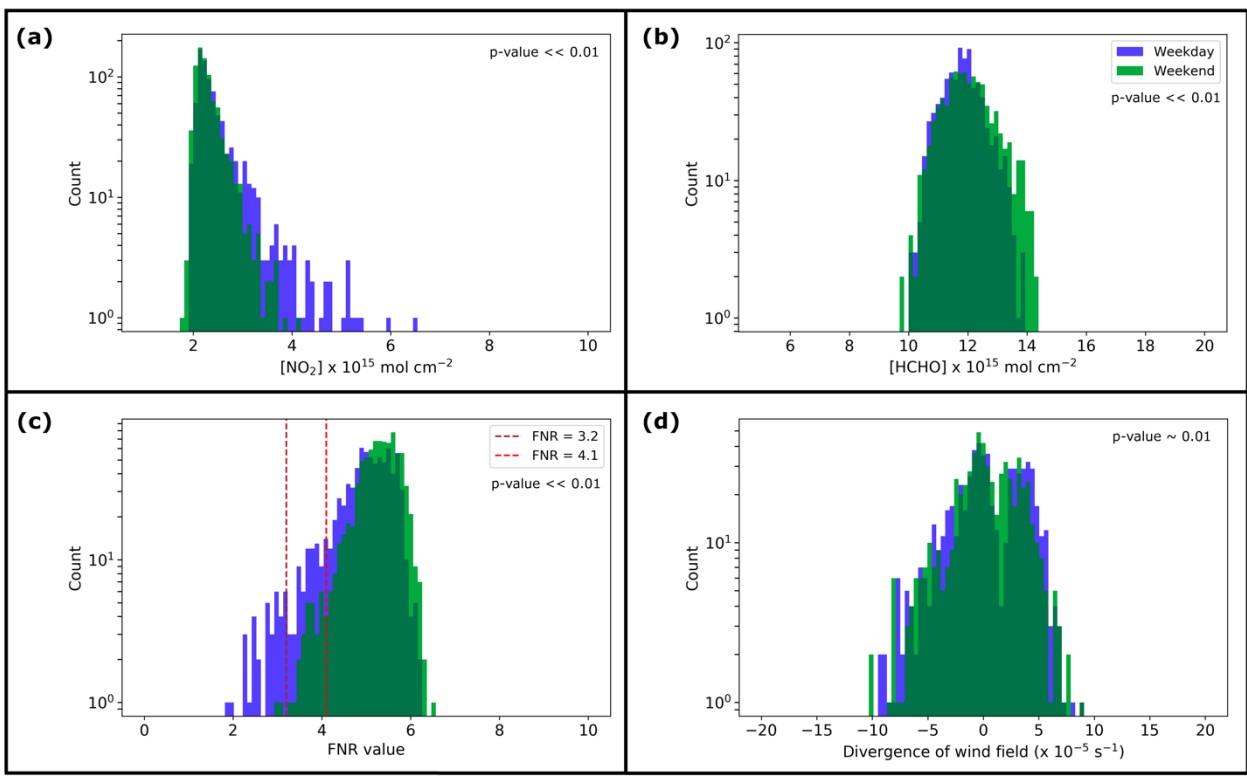

**Figure 14: Histogram plots comparing the distributions of (blue) weekday values and (green) weekend values (from Figures 9, 10, 11, and 13) of: (a) $NO_2$, (b) HCHO, (c) FNRs, and (d) wind field divergence.**



### 3.4 Limitations

The use of TROPOMI satellite retrievals of tropospheric column HCHO and $NO_2$ to indicate surface $O_3$ chemistry sensitivity

has a few limitations. First, to minimize the impacts of the lower signal-to-noise ratio of TROPOMI HCHO retrievals, we temporally aggregated the data into monthly composites. Through this process we lose the ability to detect day-to-day changes in $O_3$ chemistry. Furthermore, S5P is a sun-synchronous polar-orbiting satellite, and TROPOMI provides measurements at about 13:30 local solar time. However, $O_3$ production can occur throughout the day and its sensitivity to either $NO_x$ or VOCs can change as the atmospheric concentrations of these gases change. Higher temporal and spatial resolution satellite

measurements are needed to analyse the hourly fluctuations in $O_3$ chemistry sensitivity from space-based instruments.

Analysis of the ground data also reveals some limitations of using PAMS data. The current 24-hour long sampling procedure smooths out hourly fluctuations in HCHO levels that can occur throughout the day, which could lead to incorrect determinations of $O_3$ chemistry sensitivity when interpreting surface FNR values. Additionally, FNR threshold values can vary

from airshed to airshed due to different emissions, meteorological conditions, and other environmental factors (Lu and Chang, 1998). Higher temporal resolution HCHO surface measurements are needed at more PAMS sites to better determine $O_3$ chemistry sensitivity FNR thresholds and how they vary both diurnally and spatially. Moreover, there is a spatial disconnect when comparing TROPOMI and PAMS FNRs: the satellite FNR values (i.e., the satellite pixels composited into a grid box) cover a larger area than the point PAMS monitor measurements. Finally, TROPOMI data represent tropospheric column

densities while PAMS data represent ground-level measurements, underscoring the need to apply different $O_3$ chemistry threshold values to FNRs calculated from each source dataset.

### 4 Summary and conclusions

We assessed the ozone–$NO_x$–VOC sensitivity of the Lake Michigan region using formaldehyde to nitrogen dioxide concentration ratios ([HCHO]/[$NO_2$]; "FNRs") calculated from 2019–2021 S5P TROPOMI satellite data and 2019 PAMS

surface measurements. Results from the first part of this work showed that at PAMS Site 3103 in Schiller Park, Illinois, satellite-based FNRs were always greater in value than ground-based FNRs. This highlights the need to apply different threshold values to ground (surface) and satellite (column) FNR values when determining $O_3$ chemistry sensitivity. Specifically, we suggest applying the Jin et al. (2020; "J20") thresholds to the satellite FNRs and the Blanchard (2020) thresholds to the PAMS FNRs presented in this work.


TROPOMI FNR composites interpreted using the J20 thresholds suggest that despite increases in regional HCHO levels during Chicago $O_3$ exceedance days, the Chicago metropolitan area remains VOC-sensitive due to higher $NO_2$ levels (note that the





urban core of Chicago increases in VOC sensitivity as indicated by lower FNR values on Chicago exceedance days). Although areas surrounding the Chicago metropolitan area and north along the Lake Michigan coastline to Milwaukee, Wisconsin,

largely remain in the transition zone, the extent of the transition zone drops by 25% during exceedance days. The rest of the domain is still $NO_x$-sensitive during Chicago ozone exceedance days, but even more so (as indicated by higher FNR values) due to increased HCHO levels. Higher HCHO concentrations are likely attributed to increased temperatures during exceedance events, which leads to increased biogenic VOC emissions and increased $O_3$ production (Sillman and Samson, 1995). These results suggest that both VOC and $NO_x$ emissions controls would be necessary to decrease $O_3$ production in the Chicago

nonattainment area and for the region as a whole. Analysis of wind data shows that the typical lake breeze pattern that occurs during the TROPOMI overpass time is stronger during Chicago exceedance days, which is also likely attributable to higher land temperatures impacting this type of thermally direct circulation.

When comparing weekday and weekend TROPOMI FNR composites, there is a clear difference in $O_3$ chemistry sensitivity in

the Chicago metropolitan area. The percent of area classified as VOC-sensitive drops from 4.3 % on weekdays to 0.2 % on weekends, driven by statistically lower weekend $NO_x$ levels in the region. Similarly, the percent of area classified as having transitional $O_3$ chemistry also drops from 9.1 % on weekdays to 3.4 % on weekends due to lower $NO_x$ levels. Although the spatial distribution of differences between weekday and weekend HCHO levels are mixed between positive and negative values, the domain average has slightly higher HCHO levels on weekends. Further research is needed to investigate the possible

causes for this finding (e.g., changes in $NO_x$–VOC chemistry due to lower $NO_x$ levels on weekends). Analysis also shows no major differences between wind speed, direction, and divergence between weekdays and weekends during the TROPOMI overpass time, as expected.

The TROPOMI-based FNRs analyses in this study provide a "snapshot" of the average ozone–$NO_x$–VOC sensitivity of the

Lake Michigan region during the Sentinel-5P satellite overpass time. Although such snapshots are informative for air quality management agencies that develop $O_3$ attainment strategies, they miss the hourly fluctuations in atmospheric HCHO and $NO_2$ levels (and thus fluctuations in $O_3$ chemistry sensitivity) that occur throughout the day. The NASA Tropospheric Emissions: Monitoring of Pollution (TEMPO) mission will address such limitations by launching an instrument onboard a geostationary-orbiting satellite that is capable of measuring air pollutants, including HCHO and $NO_2$, in hourly intervals over the United

States (Zoogman et al., 2017). Other geostationary-orbiting satellite instruments such as the Geostationary Environment Monitoring Spectrometer (GEMS; Choi et al., 2018) and SENTINEL-4 (Gulde et al., 2017) will also make similar measurements as TEMPO for many countries in Asia and Europe, respectively. In the future, such instruments will provide researchers with new datasets to explore how information from satellite-based [HCHO]/[$NO_2$] ratios can be utilized to infer surface $O_3$ chemistry sensitivity at unprecedented spatiotemporal scales.



**Appendices**

**Appendix A. Chicago ozone exceedance days**

**Table A1: Dates of 2019, 2020, and 2021 O$_3$ exceedance days in the Chicago, Illinois, metropolitan area. An O$_3$ exceedance day is defined as having at least one ground monitor in the U.S. EPA Air Quality System (AQS) measuring an MDA8 O$_3$ value greater than 70 ppb.**

| 2019 Dates | Number of stations | 2020 Dates | Number of stations | 2021 Dates | Number of stations |
|---|---|---|---|---|---|
| June 5 | 2 | June 4 | 4 | May 22 | 1 |
| June 26 | 1 | June 5 | 10 | June 3 | 14 |
| June 28 | 1 | June 8 | 3 | June 4 | 5 |
| June 29 | 4 | June 16 | 2 | June 11 | 5 |
| July 3 | 2 | June 17 | 15 | June 17 | 5 |
| July 5 | 1 | June 18 | 19 | June 18 | 8 |
| July 8 | 1 | June 19 | 16 | July 20 | 5 |
| July 9 | 11 | June 27 | 2 | July 22 | 6 |
| July 13 | 1 | July 1 | 4 | July 23 | 4 |
| July 25 | 2 | July 2 | 1 | July 26 | 2 |
| August 2 | 1 | July 3 | 14 | July 27 | 1 |
| August 3 | 2 | July 6 | 13 | July 28 | 4 |
| **Total: 12 days** | | July 7 | 6 | August 4 | 3 |
| | | July 8 | 4 | August 7 | 2 |
| | | July 9 | 5 | August 10 | 1 |
| | | July 17 | 2 | August 23 | 1 |
| | | July 25 | 5 | August 25 | 6 |
| | | August 7 | 1 | August 26 | 4 |
| | | August 15 | 4 | August 27 | 4 |
| | | August 21 | 3 | September 13 | 1 |
| | | **Total: 20 days** | | October 1 | 3 |
| | | | | **Total: 21 days** | |





## Appendix B. Derivation of the uncertainty of FNR values calculated from TROPOMI retrievals


$$FNR \pm uncertainty\ in\ FNR\ value\ = \frac{[NO_2] \pm uncertainty\ in\ NO_2\ retrieval}{[HCHO] \pm uncertainty\ in\ HCHO\ retrieval}$$

$$Uncertainty\ in\ FNR\ value = \sqrt{(uncertainty\ in\ NO_2\ retrieval)^2 + (uncertainty\ in\ HCHO\ retrieval)^2}$$

Note: Based on equation for uncertainty propagation for division

| Retrieval species | Lower limit uncertainty | Upper limit uncertainty | Source(s) |
|---|---|---|---|
| NO₂ | 15 % | 50 % | Boersma et al., 2018; van Geffen et al., 2021 |
| HCHO | 30 % | 60 % | De Smedt et al., 2021 |


$$Lower\ limit\ FNR\ uncertainty = \sqrt{(0.15)^2 + (0.30)^2} \approx 0.335, or\ 33.5\ \%$$

$$Upper\ limit\ FNR\ uncertainty = \sqrt{(0.50)^2 + (0.60)^2} \approx 0.781, or\ 78.1\ \%$$





## Appendix C. TROPOMI composite statistics

**Table C1: Basic statistics for the TROPOMI typical ozone season day and Chicago ozone exceedance day composites.**

|  |  | Ozone season (OS) | Chicago exceedance days (Ex) | Difference (Ex – OS) |
|---|---|---|---|---|
| **HCHO (mol cm$^{-2}$)** | Min. | $10.2 \times 10^{15}$ | $9.64 \times 10^{15}$ | $-0.96 \times 10^{15}$ |
|  | Mean | $11.9 \times 10^{15}$ | $13.0 \times 10^{15}$ | $1.13 \times 10^{15}$ |
|  | Max. | $13.8 \times 10^{15}$ | $16.2 \times 10^{15}$ | $2.67 \times 10^{15}$ |
| **NO$_2$ (mol cm$^{-2}$)** | Min. | $1.95 \times 10^{15}$ | $1.84 \times 10^{15}$ | $-0.38 \times 10^{15}$ |
|  | Mean | $2.44 \times 10^{15}$ | $2.51 \times 10^{15}$ | $0.08 \times 10^{15}$ |
|  | Max. | $5.69 \times 10^{15}$ | $7.60 \times 10^{15}$ | $1.91 \times 10^{15}$ |
| **FNR (unitless)** | Min. | 2.12 | 1.84 | -0.53 |
|  | Mean | 4.99 | 5.34 | 0.35 |
|  | Max. | 6.12 | 7.11 | 1.54 |
| **Eastward wind (m s$^{-1}$)** | Min. | -1.50 | -2.59 | -1.41 |
|  | Mean | 0.82 | 0.78 | -0.04 |
|  | Max. | 1.93 | 2.92 | 0.64 |
| **Northward wind (m s$^{-1}$)** | Min. | -0.84 | -1.25 | -0.42 |
|  | Mean | 0.63 | 1.40 | 0.77 |
|  | Max. | 1.93 | 3.61 | 1.96 |
| **Wind divergence (s$^{-1}$)** | Min. | $-9.63 \times 10^{-5}$ | $-15.2 \times 10^{-5}$ | $-5.82 \times 10^{-5}$ |
|  | Mean | $0.41 \times 10^{-5}$ | $0.57 \times 10^{-5}$ | $0.16 \times 10^{-5}$ |
|  | Max. | $8.75 \times 10^{-5}$ | $15.1 \times 10^{-5}$ | $8.65 \times 10^{-5}$ |







665          **Table C2: Basic statistics for the TROPOMI weekday and weekend composites.**

| | | Weekday | Weekend | Difference (Weekend – Weekday) |
|---|---|---|---|---|
| **HCHO (mol cm$^{-2}$)** | **Min.** | $10.0 \times 10^{15}$ | $9.74 \times 10^{15}$ | $-1.68 \times 10^{15}$ |
| | **Mean** | $11.8 \times 10^{15}$ | $12.0 \times 10^{15}$ | $0.21 \times 10^{15}$ |
| | **Max.** | $13.9 \times 10^{15}$ | $14.3 \times 10^{15}$ | $1.93 \times 10^{15}$ |
| **NO$_2$ (mol cm$^{-2}$)** | **Min.** | $1.91 \times 10^{15}$ | $1.82 \times 10^{15}$ | $-2.60 \times 10^{15}$ |
| | **Mean** | $2.49 \times 10^{15}$ | $2.31 \times 10^{15}$ | $-0.18 \times 10^{15}$ |
| | **Max.** | $6.45 \times 10^{15}$ | $4.11 \times 10^{15}$ | $0.37 \times 10^{15}$ |
| **FNR (unitless)** | **Min.** | 1.88 | 2.94 | -1.00 |
| | **Mean** | 4.90 | 5.26 | 0.36 |
| | **Max.** | 6.23 | 6.53 | 1.56 |
| **Eastward wind (m s$^{-1}$)** | **Min.** | -1.62 | -1.35 | -0.55 |
| | **Mean** | 0.80 | 0.86 | 0.06 |
| | **Max.** | 2.85 | 2.38 | 0.56 |
| **Northward wind (m s$^{-1}$)** | **Min.** | -1.04 | -0.51 | -0.31 |
| | **Mean** | 0.50 | 0.94 | 0.43 |
| | **Max.** | 1.94 | 1.90 | 1.24 |
| **Wind divergence (s$^{-1}$)** | **Min.** | $-9.48 \times 10^{-5}$ | $-10.3 \times 10^{-5}$ | $-3.50 \times 10^{-5}$ |
| | **Mean** | $0.47 \times 10^{-5}$ | $0.28 \times 10^{-5}$ | $-0.19 \times 10^{-5}$ |
| | **Max.** | $8.71 \times 10^{-5}$ | $9.00 \times 10^{-5}$ | $2.39 \times 10^{-5}$ |




**Code & data availability**

Python scripts used to generate composites of TROPOMI data and analyse PAMS data as well as the TROPOMI composite data files we created in netCDF format are available upon request (send correspondence to acdan@wisc.edu). TROPOMI data can be downloaded from the NASA GES DISC website (https://disc.gsfc.nasa.gov/). PAMS data can be downloaded from the U.S. EPA Air Data website (https://www.epa.gov/outdoor-air-quality-data).

**Author contribution**

J. J. M. Acdan's contributions include: (1) acquiring and processing TROPOMI data, (2) processing and analysing PAMS data, (3) visualizing all data, (4) developing the methodologies, (5) interpreting the results, and (6) writing, reviewing, and editing drafts of this manuscript. R. B. Pierce's contributions include: (1) participating in the conceptualization of the study, (2) developing the methodologies, (3) supervising the project, (4) interpreting the results, and (5) reviewing and editing drafts of this manuscript. A. F. Dickens's, Z. Adelman's, and T. Nergui's contributions include: (1) conceptualizing the research
project, (2) acquiring PAMS data, (3) supervising the project, (4) reviewing and editing drafts of this manuscript, and (5) providing funding support. A. F. Dickens was additionally involved in the interpretation of the results.

**Competing interests**

The authors declare that they have no conflict of interest.

**Acknowledgements**

We acknowledge funding support from the Lake Michigan Air Directors Consortium (LADCO).






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
