# Peer review of "Examining TROPOMI formaldehyde to nitrogen dioxide ratios in the Lake Michigan region: implications for ozone exceedances"

_EGUsphere, 2022_

## Referee Comment (RC1)

Referee Comment

Title: Ozone–NO$_x$–VOC Sensitivity of the Lake Michigan Region Inferred from TROPOMI Observations and Ground-Based Measurements

General comments

Acdan et al. use TROPOMI column density retrievals and PAMS surface concentration measurements of HCHO and NO$_2$ to study HCHO, NO$_2$, and ozone production sensitivities (indicated by FNRs) in the lake Michigan region. The authors carefully composite for typical ozone season days, ozone exceedance days, weekdays, and weekends. They identify a spatial heterogeneity in ozone chemistry sensitivity for Chicago metropolitan area and its surrounding region, where the metropolitan area remains VOC-sensitive. They find that changes in FNRs on ozone exceedance days indicate an increase in NOx-sensitivity in NOx-sensitive areas and an increase in VOC-sensitivity in VOC-sensitive areas. Connecting wind fields with lake breeze provides a nice illustration of a stronger lake breeze effect on higher-ozone days.

Overall, the paper provides important implications for ozone mitigation in the Lake Michigan Region. However, a few major issues should be addressed before I recommend publication.

(1) The uncertainties in using OMI threshold values to interpret TROPOMI FNRs are briefly mentioned. However, despite similar signal-to-noise ratios, TROPOMI still shows disagreement with OMI, and it would be good to see some discussion on how this difference, along with TROPOMI bias, may affect the study results.

(2) The unresolved biases and noises in FNR would be amplified as opposed to using HCHO and NO$_2$ columns individually. The equation for uncertainty propagation was not properly implemented for division in Appendix B. For division ($z = \frac{x}{y}$): $\delta_z = \sqrt{\left(\frac{\delta_{[x]}}{[x]}\right)^2 + \left(\frac{\delta_{[y]}}{[y]}\right)^2}\,|z|$. More discussion on retrieval errors can be found in Souri et al. 2022 (https://acp.copernicus.org/preprints/acp-2022-410/).

(3) For weekday/weekend analysis, there seem to be many more days selected for weekdays (327 days) than weekends (132 days). Would the result be impacted by averaging over more days? It may be necessary to test if the same number of days were selected.

(4) It would be interesting to see how different regions are similar/different in changes on higher-ozone days. Is there any broader implication of this study on similar urban environments? For example, Tao et al. 2022 (https://pubs.acs.org/doi/full/10.1021/acs.est.2c02972) compare TROPOMI HCHO, NO$_2$, and FNRs on ozone exceedance days versus non-exceedance days and weekdays versus weekends, for summer 2018 over New York City.

Specific comments

P3. Line 89. The definition of "typical O$_3$ season days" and "exceedance days" could be moved from P3. Line 104 to here (or briefly mentioned), as they appear for the first time.

P4. Line 116-122. It is controversial to conclude that the errors affecting HCHO and $NO_2$ retrievals can be canceled out rather than amplified by using their ratio. Please see more in the general comments.

P7. Section 2.2. What are the uncertainties in the PAMS surface measurements?

P9. Line 233-235. This sentence on diurnal cycles seems confusing. Not sure how having diurnal information would make a difference in the current observations. What time of the day were the 6-day interval HCHO measurements collected? Or is it daily mean?

P12. Line 309-311. May examine the mean temperatures for each composite and verify whether the higher-ozone days co-occur with hotter temperatures.

Technical comments
P2. Line 36. Could say "exceed the NAAQS" as $O_3$ is mentioned three times in this sentence.

P6. Line 164. 15 individual monthly composites?

P6. Line 166. "Next, we  created"

---

## Author Comment (AC1)

**Response to referee #1**

**Referee Comment**

Title: Ozone–NOx–VOC Sensitivity of the Lake Michigan Region Inferred from TROPOMI Observations and Ground-Based Measurements

General comments
Acdan et al. use TROPOMI column density retrievals and PAMS surface concentration measurements of HCHO and NO2 to study HCHO, NO2, and ozone production sensitivities (indicated by FNRs) in the lake Michigan region. The authors carefully composite for typical ozone season days, ozone exceedance days, weekdays, and weekends. They identify a spatial heterogeneity in ozone chemistry sensitivity for Chicago metropolitan area and its surrounding region, where the metropolitan area remains VOC-sensitive. They find that changes in FNRs on ozone exceedance days indicate an increase in NOx-sensitivity in NOx-sensitive areas and an increase in VOC-sensitivity in VOC-sensitive areas. Connecting wind fields with lake breeze provides a nice illustration of a stronger lake breeze effect on higher-ozone days.

Overall, the paper provides important implications for ozone mitigation in the Lake Michigan Region. However, a few major issues should be addressed before I recommend publication.

**Introductory comment**

We thank referee #1 for providing thorough feedback on our manuscript. Based on both referees' comments, we have made major revisions to the paper, including:

1. Changing the title of the manuscript to "Examining TROPOMI formaldehyde to nitrogen dioxide ratios in the Lake Michigan region: implications for ozone exceedances"
2. Removing all text/figures/references/etc. relating to PAMS data
3. Re-processing the data composites, specifically:
   a. Using the reprocessed TROPOMI $NO_2$ PAL dataset so that all data come from the same processor version
   b. Removing the use of detection limit thresholds
   c. Addressing the HCHO artifact over water through a bias-correcting approach
   d. Using the same number of days for the TROPOMI weekday-weekend composites
4. Adding new 2-meter temperature composites from the NAM analysis dataset
5. Expanding on the discussions of FNR errors (e.g., citing Souri et al., 2023), the usage of the J20 thresholds, and comparisons to similar studies (e.g., Tao et al., 2022)
6. Moving some of the appendices to a supplemental information document along with new supplemental figures/tables

We believe that these changes have greatly added to the scientific content of the paper and look forward to another round of discussion, if needed.

Our responses to referee #1's specific comments are as follows:

(1) The uncertainties in using OMI threshold values to interpret TROPOMI FNRs are briefly mentioned. However, despite similar signal-to-noise ratios, TROPOMI still shows disagreement with OMI, and it would be good to see some discussion on how this difference, along with TROPOMI bias, may affect the study results.

**Response**

We agree that these topics should be addressed in the manuscript.

**Changes to manuscript**

We have added a subsection to the data & methodology section entitled "2.4 Analysis of data composites". In this subsection we note that the primary method of analysis is through taking the difference between composite categories and providing a qualitative interpretation of what that means in terms of ozone chemistry sensitivity; we can then determine if the changes are due to changes in HCHO or $NO_2$ levels, or both.

Because we use the J20 threshold values as an additional analysis tool to interpret our TROPOMI derived FNRs, we also highlight some of the disagreements between TROPOMI and OMI data, as well as assumptions and uncertainties from the J20 study, and how these factors may affect the study results.

This new subsection can be found in the revised manuscript on lines 207–223.

(2) The unresolved biases and noises in FNR would be amplified as opposed to using HCHO and NO2 columns individually. The equation for uncertainty propagation was not properly implemented for division in Appendix B. For division ($z=xy$): $\delta z=\sqrt{(\delta[x][x])2+(\delta[y][y])2}|z|$. More discussion on retrieval errors can be found in Souri et al. 2022 (https://acp.copernicus.org/preprints/acp-2022-410/).

**Response**

We thank the referee for pointing out the error in the equation.

**Changes to manuscript**

We have removed the equation from appendix section and instead added a subsection to the data & methodology section entitled "2.1.1 Errors associated with FNRs derived from S5P TROPOMI data". In this section, we reference the Souri et al. (2023) paper to provide a more detailed discussion of FNR errors and discuss what these errors imply for the FNRs we calculated for the Lake Michigan region. We reference Souri et al. (2023) eq. 15, which is the correct equation as the referee pointed out.

This new subsection can be found in the revised manuscript on lines 141–152.

(3) For weekday/weekend analysis, there seem to be many more days selected for weekdays (327 days) than weekends (132 days). Would the result be impacted by averaging over more days? It may be necessary to test if the same number of days were selected.

**Response**

We agree that it is best to test if the number of days selected impacts the results.

**Changes to manuscript**

We tested using only Tuesdays/Wednesdays in the weekday composites and Saturdays/Sundays in the weekend composites. Both composites were created with the same number of days over the 3-year period (114 days). The analysis of these new composites is largely the same as before. We replaced the old plots (created with unequal number of days) with the new weekday/weekend composites (created with equal number of days) in the revised manuscript (Figures 7–10 in the revised manuscript).

(4) It would be interesting to see how different regions are similar/different in changes on higher-ozone days. Is there any broader implication of this study on similar urban environments? For example, Tao et al. 2022 (https://pubs.acs.org/doi/full/10.1021/acs.est.2c02972) compare TROPOMI HCHO, NO2, and FNRs on ozone exceedance days versus non-exceedance days and weekdays versus weekends, for summer 2018 over New York City.

**Response**

We agree that comparisons to similar studies increases the scientific content of this paper.

**Changes to manuscript**

We added a paragraph talking about how our study is similar to Tao et al. (2022), which can be found on lines 379–383 in the revised manuscript. We end the paragraph with the following to highlight potential broader implications:

"These similarities suggest that the results presented here are broadly applicable to other coastal urban environments with $O_3$ exceedance problems. Future work could investigate FNRs over Detroit, Michigan, and Los Angeles, California, to see if this implication is true." (lines 381–383 in the revised manuscript)

**Specific comments**

P3. Line 89. The definition of "typical O3 season days" and "exceedance days" could be moved from P3. Line 104 to here (or briefly mentioned), as they appear for the first time.

**Response**

We agree with this suggestion.

**Changes to manuscript**

We moved the definitions of the "ozone season" and "exceedance days" to the introduction when the terms first appear. (line numbers 103–107 in the revised manuscript)

P4. Line 116-122. It is controversial to conclude that the errors affecting HCHO and NO2 retrievals can be canceled out rather than amplified by using their ratio. Please see more in the general comments.

**Response**

We agree with this comment.

**Changes to manuscript**

We added a subsection to the data & methodology section entitled "2.1.1 Errors associated with FNRs derived from S5P TROPOMI data" to provide a more detailed discussion of FNR errors. Please see our response to the related general comment above for more information.

P7. Section 2.2. What are the uncertainties in the PAMS surface measurements?

**Response & changes to manuscript**

To address the specific comments from Referee #2, we have decided to entirely remove the sections involving PAMS surface measurements and focus on analyzing TROPOMI and meteorological data composites.

P9. Line 233-235. This sentence on diurnal cycles seems confusing. Not sure how having diurnal information would make a difference in the current observations. What time of the day were the 6-day interval HCHO measurements collected? Or is it daily mean?

**Response & changes to manuscript**

To address the specific comments from Referee #2, we have decided to entirely remove the sections involving PAMS surface measurements and focus on analyzing TROPOMI and meteorological data composites.

P12. Line 309-311. May examine the mean temperatures for each composite and verify whether the higher-ozone days co-occur with hotter temperatures.

**Response**

We agree with this suggestion and believe it would add scientific content to the paper.

**Changes to manuscript**

We added 2-meter temperature data composite images for comparison. The data come from the NCEP North American Mesoscale (NAM) 12 km Analysis (ds609.0 | DOI: 10.5065/G4RC-1N9 | https://rda.ucar.edu/datasets/ds609.0/). These composites confirm our original suggestion that higher ozone days co-occur with hotter temperatures. These composites are Figures 2d–f and 7d–f and in the revised manuscript.

**Technical comments**

P2. Line 36. Could say "exceed the NAAQS" as O3 is mentioned three times in this sentence.

**Response**

We agree with this suggestion.

**Changes to manuscript**

We removed "$O_3$" from before NAAQS so that the sentence does not sound too repetitive.

P6. Line 164. 15 individual monthly composites?

**Response**

Originally, we created 15 individual monthly composites (May-September = 5 months) for 2019–2021 (3 years) → 5 × 3 = 15 composites. We then combined them using a weighted average to get an overall 2019-2020-2021 mean $O_3$ season composite. This was originally done to reduce processing time while creating the composites.

However, to address comments by referee #2, we decided to re-process the TROPOMI data. During this re-processing, we made our compositing script more efficient, which allowed us to create a single 2019–2020–2021 mean $O_3$ season composite. We have confirmed that taking the weighted average of individual composites and creating a single composite of all the data produce the same average values.

**Changes to manuscript**

We have removed any mention of "individual monthly composites" from the revised manuscript and updated the data & methodology section to reflect the changes in the satellite data processing.

P6. Line 166. "Next, we  created"

**Response**

We thank the referee for catching this grammatical error.

**Changes to manuscript**

We have removed the second "next" as suggested.

---

## Author Comment (AC2)

**Response to Referee #2**

Acdan et al., 2022 leveraged the satellite observations of HCHO and NO2 columns retrieved from TROPOMI radiance and a ground-based monitoring station to contrast the underlying ozone regimes in a region undergoing high ozone exceedances in different episodes such as weekday vs. weekends and ozone exceedance days vs. seasonal averaged values. They observed higher NO2 columns over Chicago during high ozone exceedances, but its dominantly VOC-sensitive regime did not change due to apparent enhancements in HCHO columns. They observed the typical weekday/weekend tendencies in the former ozone studies. The PAM measurements revealed higher FNRs than those of TROPOMI due to differences in sampling time and inherit column-to-surface discrepancies (Jin et al., 2017). Unfortunately, the scientific content of the paper is really thin; there are artifacts associated with HCHO retrievals; some assumptions about the thresholds were not well thought out; the paper does not inform about the driving factors of the PAMS vs. the satellite discrepancies, and the time period of the case study (during the lockdown) is poorly chosen. The paper also has repetitive analyzes, such as recycling the spatial distributions of HCHO and NO2 in the shape of histograms that do not provide new content (they could have been presented in SI). The paper clearly does not reach the ACP standard; thus, I recommend rejection.

**Introductory comment**

We thank referee #2 for providing thorough feedback on our manuscript. Based on both referees' comments, we have made major revisions to the paper, including:

1. Changing the title of the manuscript to "Examining TROPOMI formaldehyde to nitrogen dioxide ratios in the Lake Michigan region: implications for ozone exceedances"
2. Removing all text/figures/references/etc. relating to PAMS data
3. Re-processing the data composites, specifically:
    a. Using the reprocessed TROPOMI $NO_2$ PAL dataset so that all data come from the same processor version
    b. Removing the use of detection limit thresholds
    c. Addressing the HCHO artifact over water through a bias-correcting approach
    d. Using the same number of days for the TROPOMI weekday-weekend composites
4. Adding new 2-meter temperature composites from the NAM analysis dataset
5. Expanding on the discussions of FNR errors (e.g., citing Souri et al., 2023), the usage of the J20 thresholds, and comparisons to similar studies (e.g., Tao et al., 2022)
6. Moving some of the appendices to a supplemental information document along with new figures/tables

We believe that these changes have greatly added to the scientific content of the paper and look forward to another round of discussion, if needed.

Our responses to referee #2's specific comments are as follows:

**Major comments**

HCHO artifact: Figures 4 and 10 show elevated HCHO concentrations over Lake Michigan that are nonsensical. The surface albedo treatment in the TROPOMI HCHO retrievals most likely causes this artifact. The atmosphere cannot work in that way such that we see a sharp contrast in a relatively spatially homogenous compound like HCHO between land and water. The transport pattern shown in the draft indicates an outflow originating from the lake to the surrounding areas, so the lake will not act as a reservoir to accommodate the transported HCHO. As a result, the statistics regarding HCHO and the ratio (such as the percentage of each underlying ozone regime) are unrealistic. If the authors disagree with me, they should scientifically prove that such elevated HCHO values can prevail over the lake. Do you see the same tendency using a CTM model over the same area (e.g., https://agupubs.onlinelibrary.wiley.com/doi/full/10.1029/2022JD037042)? If yes, please break down the physiochemical processes to determine the major driver; I am very doubtful about the quality of TROPOMI HCHO over water especially lakes with complex surface albedo properties unresolved in 0.5x0.5o OMI albedo climatology used in TROPOMI HCHO retrieval.

**Response**

We agree that the higher HCHO over Lake Michigan is likely an artifact.

**Changes to manuscript**

In the manuscript, we first present the HCHO composite (Figure 4) and acknowledge the lake artifact (including discussing the fact that HCHO is a relatively well-mixed gas with no sources over the water surface and the coarse OMI surface albedo climatology used in the retrieval). Then we apply a "bias correcting" procedure by assuming that the mean HCHO over land should be equal to the mean HCHO over water. To calculate the bias, we subtract the mean over land HCHO VCD from the mean over water HCHO VCD. Finally, we subtract this bias value from all water grid box values. As a quick example, below shows Figure 4 (non-bias corrected) and Figure 5 (bias corrected):

[Figure]

The bias corrected HCHO composite for the $O_3$ season/exceedance comparison is presented in Figure 5 and the weekday-weekend comparison in Figure 9. We use the bias corrected HCHO to calculate the FNR values presented in the revised manuscript.

Figures S1/S4 and Tables S5/S6 in the supplemental information document show the bias calculations in more detail. Additionally, Figures S2/S6 show FNRs calculated using non-biased corrected HCHO for reference.

J20 assumptions: The analysis heavily relies upon the thresholds defined in J20, whose application for this case study is questionable. Two central problems exist 1) J20 thresholds are not intended for understanding the sensitivity of PO3 to NOx and VOC but rather for understanding the sensitivity of maximum peak in ambient O3 concentrations to its precursors. Ambient O3 levels can be largely impacted by physical processes such as dry deposition, transport, etc.  These two sensitivities will not be the same. J20 thresholds are case-study specific and only applicable to their time period/location because the physical processes (i.e., transport, deposition, …) can vary greatly from time to time. 2) J20 focused on OMI data possessing significant dispersions in HCHO columns (De Smedt et al., 2021) as opposed to those of TROPOMI. The spatial representation between these two sensors is also different. As the retrieval algorithm is a major source of error in the ratio, the fuzziness in J20 thresholds was induced by the errors in OMI that are largely different from those in TROPOMI. The authors must have re-calibrated J20 thresholds by establishing the same relationship between max O3 and TROPOMI HCHO and NO2 columns over their region of interest. Also, please avoid mixing up different thresholds from different studies looking at different things. For example, Schroeder et al. 2017 focused on aircraft observations that are not necessarily applicable to the columnar ratio. J20 studied ambient ozone concentrations instead of PO3. Duncan et al. 2010 used a CTM realization subjective to assumptions made for chemical mechanisms and physical processes. Comparing these numbers is apple-to-orange.

**Response**

We agree that the use of the J20 thresholds deserve more thorough discussion within the manuscript. We still believe insights can be gained from using the J20 thresholds, particularly because we are primarily interested in providing a qualitative picture of the spatial differences in ozone sensitivity between $O_3$ season days and exceedance days and identifying the causes of those changes on the most polluted days.

We believe that re-calibrating the thresholds by establishing the same relationship between high $O_3$ and TROPOMI precursors is out of the scope of this work.

**Changes to manuscript**

In the introduction section, we specify that the J20 thresholds describe sensitivity of high $O_3$ levels to precursors as opposed to $O_3$ production (lines 86–92 in the revised manuscript).

We have expanded the discussion of the use of the J20 thresholds in a newly added section called "2.4 Analysis of data composites". In this section, we mention that because the J20 thresholds describe high O3 sensitivity, they are less robust. Then we discuss some of the differences between OMI and TROPOMI. We acknowledge that these differences impact our interpretations ozone chemistry sensitivity when we apply the J20 OMI-based thresholds to TROPOMI FNRs (lines 207–223 in the revised manuscript).

Finally, we write in the limitations section that our interpretations of the changes in $O_3$ chemistry sensitivity between composite categories are best viewed through a qualitative lens (lines 489–492 in the revised manuscript).

Additionally, we have removed the PAMS surface data section and no longer refer to other thresholds specifically when talking about results (e.g., Schroeder et al. 2017, Duncan et al. 2010).

PAMS's loneliness: The authors briefly showed the contrast between the columnar observations and the surface ones in Section 3.1. They came to the conclusion that various thresholds should be used to segregate chemical conditions using satellite vs. surface observations because they saw a large offset in the PAMS FNRs. This argument is oblivious to the fact that these two datasets look at two different areas, one at the surface layer and the other one within columns, so even if we assumed a universal threshold, the underlying chemical regime would be totally different between those two regions. See Jin et al., 2017 who carefully studied the column-to-surface conversion for different areas/times. The authors could have potentially applied a conversion factor to look at the same layer. Moreover, this section is fully detached from the rest of the study. How did PAMS data look like for the weekday/weekend and ozone exceedances days/normal warm days, i.e., the rest of the paper? What can we really learn from this point measurement that TROPOMI cannot offer? Just showing the ratio difference between the surface and the column is not new; it has been carefully studied in more detail by Jin et al., 2017 and Schroeder et al., 2017 with more suitable tools and data.

**Response**

We agree that the differences between surface and column ratios are better investigated in Jin et al. (2017) and Schroeder et al. (2017). We also agree that these analyses are detached from the rest of the paper.

**Changes to manuscript**

We have decided to entirely remove the sections involving PAMS surface measurements and focus on analyzing TROPOMI and meteorological data composites.

Covid-19 time period and re-gridding: The study aimed to diagnose the chemical conditions for emission regulations; I wonder why the authors chose the covid-19 period when there were unusual disruptions in the emissions. What we can potentially learn from these ratios may not be applicable for a regular year. Also, an important advantage of using TROPOMI lies in its high spatial resolution. It is disappointing that the authors picked a 12x12 km2 resolution for their analysis, while TROPOMI offers more spatial variance within this grid.

**Response**

We chose to use the 3-year period between 2019 and 2021 so that the $O_3$ exceedance day composite was created with more data, making it more statistically robust.

We used the $12 \times 12$ km$^2$ grid based on a sensitivity test of using a $4 \times 4$ km$^2$ grid, which produced a very noisy HCHO composite and a resulting noisy FNR composite. Using the less noisy HCHO composite on the coarser grid was preferable since we wanted to assess the general spatial patterns in FNR values and ozone sensitivities.

**Changes to manuscript**

In the newly added section "2.4 Analysis of data composites", we acknowledge that the 3-year period includes years impacted by the COVID-19 pandemic (lines 207–223 in the revised manuscript). Throughout the paper, we discuss any evidence of a pandemic signal in our composites by looking at individual years (e.g., lines 286–288 and lines 441–444 in the revised manuscript; Figures S3 and S5 in the supplemental information). We do note, however, that the spatial patterns in FNR values and the associated ozone sensitivities appear generally consistent among all 3 individual years. In the summary and conclusions section, we also point to another study (Jing and Goldberg, 2022, https://doi.org/10.1016/j.apr.2021.101313) which found that meteorology (and not just $NO_x$ emissions decreases alone) explains much of the differences between $O_3$ production in Chicago in 2020 and the preceding years (lines 526–528 in the revised manuscript).

In the satellite data processing section, we provide the explanation given above for why we use a grid that is coarser than the TROPOMI pixel footprint (lines 169–171 in the revised manuscript).

The inability to explain the differences in concentrations: One of the potentially interesting tendencies observed from TROPOMI NO2 is the larger NO2 concentrations over Chicago in high ozone exceedances. This certainly deserves a more thorough discussion using EPA surface monitoring network, bottom/top-down emissions, or available CTMs. Another possible explanation that could have been easily vetted was to study the fraction of the number of weekdays/weekends for this episode. In terms of HCHO, the authors could use parametrized isoprene emissions (e.g., MEGAN) to potentially single out the biogenic contributions. There are also well-established studies performing a temperature-dependency adjustment to minimize the meteorological effect (e.g., Shen et al., 2019). Explaining tendencies adds value to the paper, not mapping out the data.

**Response**

While we agree that the suggestions above for future work would provide great context to our findings, we believe most of them are out of the scope of this specific work.

**Changes to manuscript**

For NO$_2$, we provide the following explanation for higher NO$_2$ along the shoreline between Chicago and Milwaukee during exceedance days:

"The increased NO$_2$ VCDs on exceedance days found along coastline between Milwaukee and Chicago can be partially explained by the stronger convergence of the wind field, which concentrates emissions originating in these areas along the southwestern shore of Lake Michigan." (lines 281–284 in the revised manuscript)

Additionally, we write:

"Further research is needed to determine why NO$_2$ VCDs are higher for the whole domain during exceedance days (e.g., examining emissions inventories/datasets, looking for temperature dependent natural sources of NOx, etc.)." (lines 284–286 in the revised manuscript)

For HCHO, we have added 2-meter temperature composites (Figure 2d-f) that provide evidence for our suggestion that higher temperatures lead to more biogenic HCHO emissions (and thus higher HCHO VCDs) on exceedance days.

Repeatability: The manuscript repeats the same tendencies observed from spatial distribution maps by plotting histograms which can be moved to the SI. You can briefly mention whether the differences are statistically significant in one or two sentences. This task could also be better executed by taking a different part of the distribution, like what was done beautifully by Lin et al., 2015 (https://www.nature.com/articles/ncomms8105). In general, two things can degrade the quality of a paper: i) repeating what other people have already done and ii) repeating the same results with a different presentation (aka fillers). There are many aspects pertaining to the analysis that deserves deeper analysis. More in-depth studies can be found related to this region's ratio and chemistry (e.g., Abdi-Oskouei et al.).

**Response**

We thank referee #2 for their suggestions regarding the K-S testing and histogram plots.

**Changes to manuscript**

We have removed all histogram plots from the manuscript. Additionally, we have adopted the K-S testing procedure by Lin et al. (2015) as suggested. We added a subsection called "2.4.1 Significance testing" to describe the methodology (lines 224–232 in the revised manuscript). We describe the K-S test results for each variable/composite in their relevant sections and provide a summary of the results in Tables S3 and S4 in the supplemental information.

**Specific Comments:**

L50. You mentioned two regimes, but you will define three ones.

**Response**

We thank referee #2 for pointing out this mistake.

**Changes to manuscript**

We restructured the introduction paragraphs talking about ozone chemistry regimes. Additionally, we added more details regarding ozone production in general to address this comment and many of the following comments as well. The paragraph in which the regimes are discussed can be found on lines 50–60 in the revised manuscript.

L53. HO2 needs to be accounted too.

**Response**

We agree that more information should be provided regarding the ozone production chain reactions.

**Changes to manuscript**

We included more details about ozone production reactions in the introduction, including those involving $HO_2$. This can be found on lines 43–48 in the revised manuscript.

L54. What type of non-linear chemistry? Please elaborate.

**Response**

We agree that we can elaborate further.

**Changes to manuscript**

We included more details about ozone production reactions and chemistry sensitivities in the introduction. This can be found on lines 43–60 in the revised manuscript.

Line 54-55. The definition of NOx-sensitive or VOC-sensitive regimes is irrelevant to the availability of free oxygen atoms. In NOx-sensitive conditions, PO3 is reduced due to decreased [NO][RO2] and [NO][HO2] because all terms are reduced. [RO2] and [HO2] are efficiently removed in NOx-sensitive conditions, yielding H2O2. In rich NOx regions, so much NOx is available that terminates OH/HO2 cycling (the ROx cycle) through NO2+OH. You need to involve the ROx-HOx cycle in this paragraph. It may also be advantageous to talk about OPEs (how much O3 is produced per NOx molecule), which vary from NOx-sensitive (high OPE) to VOC-sensitive (low OPEs) conditions.

**Response**

We agree that we should be more descriptive when talking about ozone production chemistry sensitivities.

**Changes to manuscript**

We included more details about ozone production reactions and chemistry sensitivities in the introduction. This can be found on lines 43–60 in the revised manuscript. However, we do not talk about OPEs because we do not want to make the introduction section too long.

L55-60. Jin and Holloway, 2015 are not the founders of chemical condition labels. Please use a better reference, such as Sillman et al., 2002 or Duncan et al., 2011.

**Response**

We thank referee #2 for this suggestion.

**Changes to manuscript**

We adjusted the references for the chemical sensitivity labels, which can be found on lines 50–60 in the revised manuscript.

L61-62. But didn't he conclude that H2O2/HNO3 was the most viable indicator fully describing the HOx-ROx cycle?

**Response**

Yes, Sillman (1995) did conclude that $H_2O_2/HNO_3$ was one of the most robust indicator ratios.

**Changes to manuscript**

We added a discussion referencing the above fact, but also acknowledging that $H_2O_2$ and $HNO_3$ levels/VCDs are not regularly measured/observed. We then transition to talking about $HCHO/NO_2$, the indicator we use in this work. These changes can be found on lines 69–78 in the revised manuscript.

L62. HCHO is not a proxy for VOC concentrations. It is a proxy for VOC reactivity.

**Response**

We thank referee #2 for pointing out this wrong use of terminology.

**Changes to manuscript**

We have changed the wording to say "VOC reactivity" on line 74.

L68. We shouldn't rule out the importance of H2O2/HNO3.

| **Response** |
| --- |
| We thank referee #2 for bringing up the importance of $H_2O_2$/$HNO_3$ |
| **Changes to manuscript** |
| We added an additional discussion referencing the above fact, but also acknowledging that $H_2O_2$ and $HNO_3$ levels/VCDs are not regularly measured/observed. We then transition to talking about HCHO and $NO_2$ since they are measurable from space. These changes can be found on lines 69–78 in the revised manuscript. |

L67. But NOy can provide information on how transported NOx from far areas can affect local PO3. I don't think it's necessarily a weakness.

| **Response** |
| --- |
| We thank referee #2 for pointing this out. |
| **Changes to manuscript** |
| We have changed the phrasing from "more useful" to "another useful" so that it does not seem like we are suggesting that HCHO/$NO_y$ is not a useful indicator:

"Building upon Sillman's work, Tonnesen and Dennis (2000) found that HCHO/NO2 ("FNR" for the rest of this paper) is **another useful indicator** of ozone–$NO_x$–VOC sensitivity since HCHO and $NO_2$ have similar lifetimes (on the order of hours)." (lines 75–77 in the revised manuscript) |

L79-81. What do you mean by avoiding? They ignored the critical fact that PO3 is not equal to O3. O3 can easily get impacted by meteorology and dry deposition, which are not informed by the ratio. Please rewrite this part.

| **Response** |
| --- |
| We agree that further discussion about the J20 study and the use of thresholds in our study is warranted. |
| **Changes to manuscript** |
| Please see our response to referee #2's major comment about the J20 thresholds above for the changes we made to the manuscript. |

Table1. These thresholds do not define the regimes you defined earlier. They are not directly related to PO3. What is the definition of VOC-sensitive from an ambient O3 concentration perspective? You should carefully describe the assumption J20 made and its major limitations.

**Response**

We agree that further discussion about the J20 study and the use of thresholds in our study is warranted.

**Changes to manuscript**

Please see our response to referee #2's major comment about the J20 thresholds above for the changes we made to the manuscript. Additionally, we have changed the table caption to say: "J20 FNR threshold values indicating different **high O$_3$** chemistry sensitivities for Chicago, Illinois, U.S." so as to not confuse this with PO$_3$. (line 96 in the revised manuscript)

L85. This time period is during the lockdown. How informative is the case study for a normal year?

**Response**

We believe that our results are still applicable to other years. As mentioned above in our response to the major comment, the same general pattern of FNR values and inferred ozone chemistry sensitivities is seen in individual composites for 2019, 2020, and 2021.

**Changes to manuscript**

We highlight any differences in the composites among the individual years. Please see our response to referee #2's major comment about the COVID-19 period for more details.

L101. Why do you need both versions?

| **Response** |
| --- |
| We thank referee #2 for this question; we have re-made the composites (see below). |
| **Changes to manuscript** |
| We re-made our composites using S5P PAL TROPOMI $NO_2$ data (https://data-portal.s5p-pal.com/products/no2.html), which is a harmonized dataset for $NO_2$ from 2018–2021 using the same processor version (thus removing the discontinuity).

We still had to use V1 and V2 of HCHO data product because no harmonized HCHO product exists that contains our entire study period (processor changed version in July 2020). However, the changes between versions of the HCHO product are not as drastic as the changes for $NO_2$. |

L115. Errors in AMFs also contribute to the total error.

| **Response** |
| --- |
| We thank referee #2 for pointing this out. |
| **Changes to manuscript** |
| We have removed the end of the sentence so that it states:

"The total uncertainty in HCHO tropospheric vertical column density retrievals is currently estimated to be between 30–60 % in polluted conditions". (lines 137–138 in the revised manuscript) |

L120. What assumption did they make to say that? The surface albedo and aerosol effects can vary between 340 and 440 nm.

**Response**

We thank referee #2 for the question.

**Changes to manuscript**

We have revised our discussion of errors by adding a subsection to the data & methodology section entitled "2.1.1 Errors associated with FNRs derived from S5P TROPOMI data". In this section, we reference Souri et al. (2023) to provide a more detailed discussion of FNR errors and discuss what these errors imply for the FNRs we calculated for the Lake Michigan region.

This new subsection can be found in the revised manuscript on lines 141–152.

L121. The correlated term should be "-2cov(HCHO, NO2)/(HCHO×NO2)". So if HCHO and NO2 retrievals are positively correlated, they will only reduce the total relative errors when either NO2 or HCHO values are low. The correlated term will likely be small in polluted areas where HCHO and NO2 are elevated.

**Response**

We thank referee #2 for pointing this out.

**Changes to manuscript**

We have revised our discussion of errors by adding a subsection to the data & methodology section entitled "2.1.1 Errors associated with FNRs derived from S5P TROPOMI data". In this section, we reference Souri et al. (2023) to provide a more detailed discussion of FNR errors and discuss what these errors imply for the FNRs we calculated for the Lake Michigan region.

This new subsection can be found in the revised manuscript on lines 141–152.

L128. I am not sure if I agree with the discussion about SNR. SNR has a specific definition related to the instrument specifications and the observed radiance. HCHO retrieval is inherently inferior because its optical depth (despite being higher than NO2) is located in the UV range where Rayleigh scattering and O3 absorption prevail, resulting in a less robust spectral fitting.
* * *
**Response**

We agree with this comment.

**Changes to manuscript**

We have re-worded the phrasing to be more accurate & specific:

"However, HCHO has an optical density that is an order of magnitude smaller than that of $NO_2$ because the spectral band its retrieval is derived from is in the UV range where Rayleigh scattering and ozone absorption occur (De Smedt et al., 2018). As a result, individual HCHO retrievals are noisier than $NO_2$ retrievals." (lines 156–159 in the revised manuscript)
* * *
L135. The detection limit is sensor/retrieval specific; those studies are not applicable. Why not use TROPOMI studies? De Smedt et al. 2021 say $3 \times 10^{15}$ molec.cm$^{-2}$ for TROPOMI, which is an improvement of a factor of 2 compared to OMI.
* * *
**Response**

We thank referee #2 for pointing this out. However, we no longer view detection limit filters as necessary because we are compositing the data on longer time scales, and other studies do not employ them when using TROPOMI data (e.g., see
https://pubs.acs.org/doi/full/10.1021/acs.est.2c02972,
https://agupubs.onlinelibrary.wiley.com/doi/full/10.1029/2020EF001665,
https://acp.copernicus.org/articles/23/1963/2023/)

Furthermore, the minimum value of all HCHO composites presented is greater than the $3 \times 10^{15}$ molec.cm$^{-2}$ detection limit reported by De Smedt et al. (2021). To our knowledge, no published paper exists that reports a detection limit for TROPOMI $NO_2$ tropospheric vertical column density.

**Changes to manuscript**

We removed the use of detection limit filters during the re-processing of the data.

L135. Also, I am unsure if I agree that the SNR is the same between OMI and TROPOMI. What does the literature say? When comparing SNRs, we should account for the footprint, so you have to normalize it by pixel size.

**Response**

We agree that comparing SNRs should consider footprint size.

**Changes to manuscript**

Because we are no longer using detection limit thresholds, we remove any mentions of OMI and TROPOMI having similar SNRs.

L154. Why do you degrade TROPOMI spatial variance by upscaling it to 12x12 km2 when it provides higher spatial information?

**Response**

We thank referee #2 for the question.

**Changes to manuscript**

Please see our response the major comment about the choice of using a coarser grid.

L174. What are the weights? The spatial response function?

**Response**

We thank referee #2 for these questions.

**Changes to manuscript**

During the re-processing of our data, we made our compositing script more efficient, which allowed us to create singular 2019–2020–2021 mean composites as opposed to monthly/yearly ones. We have confirmed that taking the weighted average of individual composites and creating a single composite of all the data produce the same average values.

Section 2.2. Please provide the errors associated with PAM measurements. Also, because TROPOMI captures one snapshot, can we rely on monthly-averaged samples from in-situ measurements? Large diurnal variability is associated with HCHO and NO2, which is not resolved in PAMS.

**Response**

We thank referee #2 for the question and suggestion.

**Changes to manuscript**

As mentioned in our response to the major comment above, we have removed all sections regarding PAMs measurements.

L258. Those thresholds are not necessarily related to satellites. So I don't think you should put all of them in one basket.

**Response**

We thank referee #2 for the suggestion.

**Changes to manuscript**

As mentioned in our response to the major comment above, we have removed all sections regarding PAMs measurements.

L289. Some hypotheses based on previous works?

**Response**

We thank referee #2 for the question.

**Changes to manuscript**

Please see our response to the major comment regarding higher $NO_2$ on exceedance days.

L311. Does an increase in biogenic VOC always lead to higher O3? I think you are trying to say here about the relationship between O3 and increased temperature. See Figure 8 at https://pubs.acs.org/doi/full/10.1021/cr5006815. You shouldn't rule out the effect of RO2NO2. Can you show the 2m air temperature difference too?

**Response**

We are not saying that increases in biogenic VOC always lead to higher $O_3$. We are pointing out that on Chicago exceedance days, TROPOMI HCHO VCDs are higher, suggesting higher biogenic VOC emissions.

**Changes to manuscript**

We rephrased the sentence to:

"Because positive differences occur over the entire domain, the higher HCHO abundances are likely due to increased temperatures during $O_3$ exceedance events (**Fig. 2f**), which lead to increased biogenic VOC emissions and thus increased $O_3$ production in regions with VOC-sensitive chemistry (Sillman and Samson, 1995)." (lines 327–330 in the revised manuscript)

We provided 2-meter temperature composites as Figures 2d–f.

L365. This is a generic tendency you will observe in any city worldwide. As NOx dilutes far from the sources, the chemical condition becomes less VOC-sensitive.

**Response**

We thank referee #2 for the comment.

**Changes to manuscript**

We have not made any changes based on this comment.

L409. I don't understand the connection between HCHO and thermal gradients. Why don't we look into air temperature from a model?

**Response**

We thank referee #2 for the question/suggestion.

**Changes to manuscript**

We provided 2-meter temperature composites as Figures 2d–f.

L410-414. If this is true, why is HCHO larger over the lake than the land? See my major comment.

**Response**

We thank referee #2 for the question.

**Changes to manuscript**

Please see our response to the major comment regarding the over water HCHO artifact.

Figure 8. I'm surprised by the KS test saying that the distributions of NO2 are statistically different. How many times have the tests been done? Are they done on the total distribution or a specific part of it? Please see the analysis nicely done at https://www.nature.com/articles/ncomms8105. I really don't see them being too different.

**Response**

We thank referee #2 for the questions and suggestion.

**Changes to manuscript**

We have removed all histogram plots from the manuscript. Additionally, we have adopted the K-S testing procedure by Lin et al. (2015) as suggested. We added a subsection called "2.4.1 Significance testing" to describe the methodology (lines 224–232 in the revised manuscript). We describe the K-S test results for each variable/composite in their relevant sections and provide a summary of the results in Tables S3 and S4 in the supplemental information. Our new results indicate a significant difference in $NO_2$ VCDs between the $O_3$ season and exceedance days.

Figure8. What do we learn from these histograms that were not presented in the previous plots? I feel like the authors repeat the same tendencies. It really doesn't add new information.

**Response**

We thank referee #2 for this comment.

**Changes to manuscript**

We have removed all histogram plots from the manuscript.

L465. This is too speculative, given the HCHO artifact. Also, how sure are we that isoprene emissions behave similarly in two episodes?

**Response**

We thank referee #2 for this comment.

**Changes to manuscript**

Our new results (Figure 9) clearly show higher HCHO over the land and water in the southern part of the domain on weekends. In addition to providing a hypothesis for why this might be happening, we say future research is needed to find causes for our finding here.

L563. What do you mean by saying that ozone production occurs throughout the day? There is no production at nighttime.

**Response**

We meant to say that $O_3$ sensitivity to precursors can change hourly, which is not captured by the once daily data provided by TROPOMI.

**Changes to manuscript**

We have rephrased the sentence to say:

"However, the sensitivity of $O_3$ levels to $NO_x$ and VOCs can change as the atmospheric concentrations of these gases change on shorter timescales (e.g., hourly)." (lines 483–485 in the revised manuscript).

Last paragraph in conclusion: Please always provide aspects that your analysis has focused on. Your study did not quantify the temporal representation errors to gauge the importance of TROPOMI vs. GEO satellites. This paragraph is just a filler with no relevance to the results.

**Response**

We thank referee #2 for the comment. While we agree that we did not quantify the temporal representation errors to gauge the importance of TROPOMI vs. GEO satellites, we believe it is important to mention the how the upcoming geostationary satellites will provide future opportunities to conduct FNR research with new datasets.

**Changes to manuscript**

We have shortened this last paragraph to:

"Future geostationary satellite instruments, such as the NASA Tropospheric Emissions: Monitoring of Pollution (TEMPO) set to launch in 2023 (Zoogman et al., 2017) and the ESA SENTINEL-4 set to launch in 2024 (Gulde et al., 2017), will make measurements of HCHO and $NO_2$ in hourly intervals over the United States and Europe, respectively. The datasets produced by these instruments will provide researchers with new opportunities to explore the viability of using satellite-derived FNRs to infer surface ozone–$NO_x$–VOC sensitivity at unprecedented spatiotemporal scales." (lines 530–534 in the revised manuscript)

**Editorial Comments:**

L33. Longer than what?

**Response**

We thank referee #2 for the question.

**Changes to manuscript**

We have rephrased the sentence to say:

"Acute exposure to elevated $O_3$ levels can cause respiratory problems (e.g., asthma attacks) while chronic exposure can lead to premature death from respiratory and circulatory system illnesses…" (lines 32–34 in the revised manuscript)

L116. Please use the right symbol for times instead of x.

**Response**

We thank referee #2 for the suggestion.

**Changes to manuscript**

We have replaced all instances of "x" with the correct symbol "×".

L117. Molec. is better over mol. Please remake all figures and apply this to the text. Mol can be wrongly interpreted as mole.

**Response**

We thank the referee for the suggestion.

**Changes to manuscript**

We have replaced all instances of "mol" with the correct symbol "molec."

Appendixes could be moved to SI.

**Response**

We thank the referee for the suggestion.

**Changes to manuscript**

We have removed all appendices from the revised manuscript and created a new supplemental information document.

---

## Author Comment (AC3)

**Table S1.** Dates of 2019, 2020, and 2021 $O_3$ exceedance days in the Chicago-Naperville-Elgin, Illinois-Indiana-Wisconsin, core-based statistical area (hereafter the Chicago metropolitan area, or "CMA" for short). An $O_3$ exceedance day is defined as having at least one ground monitor in the U.S. EPA Air Quality System (AQS) measuring a maximum daily 8-hour average (MDA8) $O_3$ value greater than 70 parts per billion (ppb). The "# stations" column indicates how many monitors in the CMA measured an exceedance.

**2019**

| Dates | # stations | Day of week |
|---|---|---|
| 5-Jun | 4 | Wed |
| 26-Jun | 1 | Wed |
| 28-Jun | 1 | Fri |
| 29-Jun | 6 | Sat |
| 1-Jul | 1 | Mon |
| 3-Jul | 3 | Wed |
| 5-Jul | 1 | Fri |
| 8-Jul | 1 | Mon |
| 9-Jul | 12 | Tue |
| 13-Jul | 1 | Sat |
| 25-Jul | 2 | Thu |
| 2-Aug | 2 | Fri |
| 3-Aug | 3 | Sat |
| Total: 13 days | | |
| 10 weekdays, 3 weekends | | |

**2020**

| Dates | # stations | Day of week |
|---|---|---|
| 4-Jun | 5 | Thu |
| 5-Jun | 10 | Fri |
| 8-Jun | 3 | Mon |
| 16-Jun | 2 | Tue |
| 17-Jun | 15 | Wed |
| 18-Jun | 20 | Thu |
| 19-Jun | 16 | Fri |
| 27-Jun | 2 | Sat |
| 1-Jul | 4 | Wed |
| 2-Jul | 1 | Thu |
| 3-Jul | 14 | Fri |
| 6-Jul | 14 | Mon |
| 7-Jul | 6 | Tue |
| 8-Jul | 3 | Wed |
| 9-Jul | 4 | Thu |
| 17-Jul | 2 | Fri |
| 25-Jul | 5 | Sat |
| 7-Aug | 1 | Fri |
| 15-Aug | 4 | Sat |
| 21-Aug | 3 | Fri |
| Total: 20 days | | |
| 17 weekdays, 3 weekends | | |

**2021**

| Dates | # stations | Day of week |
|---|---|---|
| 22-May | 1 | Sat |
| 3-Jun | 14 | Thu |
| 4-Jun | 5 | Fri |
| 11-Jun | 5 | Fri |
| 17-Jun | 5 | Thu |
| 18-Jun | 8 | Fri |
| 20-Jul | 5 | Tue |
| 22-Jul | 6 | Thu |
| 23-Jul | 4 | Fri |
| 26-Jul | 2 | Mon |
| 27-Jul | 1 | Tue |
| 28-Jul | 5 | Wed |
| 4-Aug | 3 | Wed |
| 7-Aug | 2 | Sat |
| 23-Aug | 1 | Mon |
| 25-Aug | 6 | Wed |
| 26-Aug | 4 | Thu |
| 27-Aug | 4 | Fri |
| 13-Sep | 1 | Mon |
| 1-Oct | 3 | Fri |
| Total: 20 days | | |
| 18 weekdays, 2 weekends | | |

**Table S2.** Distribution of the number of CMA $O_3$ exceedance days by year and day of week. The last two columns report the percentage of exceedances that occurred on weekdays and weekends for that row's time period.

| | CMA ozone exceedance day of week distribution | | | | | | | | | |
|---|---|---|---|---|---|---|---|---|---|---|
| Year(s) | Monday | Tuesday | Wednesday | Thursday | Friday | Saturday | Sunday | All days | Weekdays | Weekends |
| 2019 | 2 | 1 | 3 | 1 | 3 | 3 | 0 | 13 | 77 % | 23 % |
| 2020 | 2 | 2 | 3 | 4 | 6 | 3 | 0 | 20 | 85 % | 15 % |
| 2021 | 3 | 2 | 3 | 4 | 6 | 2 | 0 | 20 | 90 % | 10 % |
| All years | 7 | 5 | 9 | 9 | 15 | 8 | 0 | 53 | 85 % | 15 % |

**Table S3.** Kolmogorov-Smirnov (K-S) test results for comparison of the ozone season and CMA exceedance days composites. For each variable, 1000 K-S tests were performed using a random subsampling approach at the 98 % confidence level. In the "median p-value" column, values <0.001 are represented as 0, and bolded values indicate overall statistically significant results (median p-value <0.020). The "wind divergence – tails" variable refers to only subsampling values greater or less than one standard deviation from the mean wind divergence value. *Sig. diff. = significant difference

| 2-subsample K-S test results: ozone season vs. CMA exceedance days | | | | |
|---|---|---|---|---|
| Variable | Percentage of tests with a *sig. diff. | Median p-value | Full sample size | Subsample size |
| HCHO bias corrected | 100 % | **0.000** | 1146 | 286 |
| NO$_2$ | 100 % | **0.000** | 1146 | 286 |
| FNR bias corrected | 100 % | **0.000** | 1146 | 286 |
| Wind divergence | 1.8 % | 0.117 | 880 | 220 |
| Wind divergence – tails | 97.1 % | **0.001** | 241 | 60 |
| 2-m temperature | 100 % | **0.000** | 1242 | 310 |

**Table S4.** Kolmogorov-Smirnov (K-S) test results for comparison of the weekday and weekend composites. For each variable, 1000 K-S tests were performed using a random subsampling approach at the 98 % confidence level. In the "median p-value" column, values <0.001 are represented as 0, and bolded values indicate overall statistically significant results (median p-value <0.020). The "wind divergence – tails" variable refers to only subsampling values greater or less than one standard deviation from the mean wind divergence value. *Sig. diff. = significant difference

| 2-subsample K-S test results: weekdays vs. weekends | | | | |
|---|---|---|---|---|
| Variable | Percentage of tests with a *sig. diff. | Median p-value | Full sample size | Subsample size |
| HCHO bias corrected | 100 % | **0.000** | 1146 | 286 |
| NO$_2$ | 71.9 % | **0.013** | 1146 | 286 |
| FNR bias corrected | 100 % | **0.000** | 1146 | 286 |
| Wind divergence | 0 % | 0.607 | 880 | 220 |
| Wind divergence – tails | 0 % | 0.821 | 241 | 60 |
| 2-m temperature | 65.5 % | 0.023 | 1242 | 310 |

[Figure]

**Figure S1.** Boxplot distributions of TROPOMI HCHO vertical column densities (VCDs) for the $O_3$ season and CMA exceedance day composites separated by over land and over water values. The mean of each distribution is represented by an orange horizontal line. The difference in means between the over land and over water values (Δ) is the "over water bias value".

**Table S5.** HCHO absolute and relative over water bias values for the $O_3$ season and CMA exceedance day composites. The absolute over water bias value is calculated by subtracting the mean over HCHO land value from the mean HCHO over water value. The relative over water bias value is calculated as $(\frac{HCHO_{water\_mean} - HCHO_{land\_mean}}{HCHO_{land\_mean}}) \times 100\ \%$.

| Composite category | Absolute over water bias | Relative over water bias |
|:---:|:---:|:---:|
| Ozone season | $1.1083 \times 10^{15}$ molec. cm$^{-2}$ | +14.05 % |
| CMA exceedance days | $1.6143 \times 10^{15}$ molec. cm$^{-2}$ | +15.51 % |

[Figure]

**Figure S2.** TROPOMI-derived 2019–2021 FNR values (calculated using non-bias corrected HCHO values) in the Lake Michigan region during: (a) the ozone season (OS), (b) CMA exceedance days (Ex), and (c) the difference between them (Ex – OS). Jin et al. (2020; "J20") threshold interpretation of 2019–2021 ozone chemistry sensitivity (using non-bias corrected FNR values) during: (d) the ozone season, (e) CMA exceedance days, and (f) the percent of the domain area classified as each J20 sensitivity regime. Mean 10-meter winds are represented by arrows.

[Figure]

**Figure S3.** TROPOMI-derived composites of mean weekday tropospheric $NO_2$ VCDs in the Lake Michigan region during: (a) 2019, (b) 2020, and (c) 2021. The difference between weekday and weekend tropospheric $NO_2$ VCDs in the Lake Michigan region during: (d) 2019, (e) 2020, and (f) 2021. Mean 10-meter winds are represented by arrows.

[Figure]

**Figure S4.** Boxplot distributions of TROPOMI HCHO VCDs for the weekday and weekend composites separated by over land and over water values. The mean of each distribution is represented by an orange horizontal line. The difference in means between the over land and over water values (Δ) is the "over water bias value".

**Table S6.** HCHO absolute and relative over water bias values for the weekday and weekend composites. The absolute over water bias value is calculated by subtracting the mean over HCHO land value from the mean HCHO over water value. The relative over water bias value is calculated as $(\frac{HCHO_{water\_mean} - HCHO_{land\_mean}}{HCHO_{land\_mean}}) \times 100\,\%$.

| Composite category | Absolute over water bias | Relative over water bias |
|:---:|:---:|:---:|
| Weekday | $1.2589 \times 10^{15}$ molec. cm$^{-2}$ | +17.2 % |
| Weekend | $1.0029 \times 10^{15}$ molec. cm$^{-2}$ | +12.7 % |

[Figure]

**Figure S5.** TROPOMI-derived composites of mean weekend tropospheric HCHO VCDs in the Lake Michigan region during: (a) 2019, (b) 2020, and (c) 2021. The difference between weekday and weekend tropospheric HCHO VCDs in the Lake Michigan region during: (d) 2019, (e) 2020, and (f) 2021. Mean 10-meter winds are represented by arrows.

[Figure]

**Figure S6.** TROPOMI-derived 2019–2021 FNR values (calculated using non-bias corrected HCHO values) in the Lake Michigan region during: (a) weekdays, (b) weekends, and (c) the difference between them (weekend – weekday). J20 threshold interpretation of 2019–2021 ozone chemistry sensitivity (using non-bias corrected FNR values) during: (d) weekdays, (e) weekends, and (f) the percent of the domain area classified as each J20 sensitivity regime. Mean 10-meter winds are represented by arrows.

---

## Author Response (AR2)

**Final author reply to editor (second revision)**

**Dear handling editor:**

Based on both referees' comments we have made the following new revisions to the manuscript:

1. Adding an analysis to estimate how much of the differences between the ozone season and exceedance day composites are due to intra-seasonal variations versus episodic changes on exceedance days,
2. Expanding discussions of: (a) possible satellite retrieval biases due to vertical profile and temperature differences on exceedance days, (b) the possible impact of the coarse albedo climatology dataset on the TROPOMI high HCHO artifact over the Great Lakes, and (c) "noise" levels of 4 km and 12 km versions of the TROPOMI composites,
3. Including a comparison of AQS surface $NO_2$ measurements to our TROPOMI-based results, and
4. Emphasizing new insights gained from and the value of this study in the abstract and the conclusions sections.

In the "tracked changes" version of the manuscript, we highlight changes/additions in red text. Our responses to the individual referee comments begin on the next page.

**Response to referee #3**

**Referee's introductory comment**

In this paper, Acdan et al. analyzed the TROPOMI HCHO and NO2 VCDs and their ratio (FNR) over the Lake Michigan region. They compared 3-year (2019-2021) composite meteorology, HCHO, NO2, and FNR maps between days with ozone exceedance and other days during the ozone season (May to September). They found that on ozone exceedance days, HCHO, NO2, and FNR tend to be greater over the region, and lake breeze circulation also tends to be stronger. This points to the importance of meteorology in ozone pollution episodes in the area. Similar comparisons were also made between weekdays and weekends. Overall, this paper demonstrates the application of TROPOMI data in regional air quality study and should be of interest to the readers of ACP. The results are largely qualitative and subject to limitations owing to uncertainties in the TROPOMI data products. The authors addressed some of the comments from both reviewers, but there are still major concerns (see specific comments below) about the data analysis method and the robustness of the results presented. I feel that major revisions are necessary before the paper can be considered for publication in ACP.

**Response to introductory comment**

We thank referee #3 for providing thorough feedback on our revised manuscript. Based on both referees' comments, we have made the following new revisions to the manuscript:

1. Adding an analysis to estimate how much of the differences between the ozone season and exceedance day composites are due to intra-seasonal variations versus episodic changes on exceedance days,
2. Expanding discussions of: (a) possible satellite retrieval biases due to vertical profile and temperature differences on exceedance days, (b) the possible impact of the coarse albedo climatology dataset on the TROPOMI high HCHO artifact over the Great Lakes, and (c) "noise" levels of 4 km and 12 km versions of the TROPOMI composites,
3. Including a comparison of AQS surface $NO_2$ measurements to our TROPOMI-based results, and
4. Emphasizing new insights gained from and the value of this study in the abstract and the conclusions sections.

Our responses to referee #3's specific comments are as follows:

**Referee's specific comments**

(1) Most of the O3 exceedance days are in June and July as shown in the supplemental material, whereas the non-O3 pollution days are probably more evenly distributed from May to September (give that >450 days were used for the composite). One may argue that the differences between the two composites in meteorology and chemical composition can well be due to seasonal changes in meteorology, emissions, and chemical processes (as well as seasonal changes in TROPOMI retrieval performance). How do you separate the effects of seasonal changes vs. episodic events?

**Response**

This is a very insightful comment, and we agree that it is important to separate the effects of intra-seasonal changes and episodic events.

Because 94 % of ozone exceedance events occur in June, July, and August (**Table S1**) in the Chicago metropolitan area (CMA), it is possible that the differences we see between the ozone season and CMA exceedance day composites are due to intra-seasonal changes. More specifically, the inclusion of May and September data in the ozone season composite may be the cause behind the composite differences because the data used in the exceedance day composites mostly come from June–August TROPOMI observations.

To estimate the effect of intra-seasonal changes, we created boxplot distributions by month for TROPOMI $NO_2$ and HCHO composite values and NAM temperature composite values. Then we compared the June–August mean values (when most exceedances occur) to the May–September mean values (entire ozone season). We deemed the difference between the June–August mean and May–September mean the amount of change we expect to see in our difference composites (**Figs. 2f, 3c, and 5c**) due to intra-seasonal changes. Finally, we compared this difference to the mean difference we see in the main text difference composites (**Figs. 2f, 3c, and 5c**). Dividing these two values gives us an estimate of how much (%) of the change shown in our main text figures are due to intra-seasonal differences, while the remaining amount of difference we prescribe as due to changes in environmental conditions on exceedance days.

**Changes to manuscript**

In the main text, we added the following discussion to Section 3.2.1 on lines 406–419:

It is important to discuss here whether the differences between the $O_3$ season and CMA exceedance day composites are caused by intra-seasonal changes or episodic changes inherent to $O_3$ exceedance days. Because 94 % of ozone exceedance days occur in June, July, and August (**Table S1**) in the Chicago metropolitan area (CMA), it is possible that the differences we see between the ozone season and CMA exceedance day composites are due to intra-seasonal changes. More specifically, the inclusion of

May and September data in the ozone season composite may be the cause of the composite differences because the data used in the exceedance day composites mostly come from June–August TROPOMI observations. To estimate the effects of intra-seasonal changes, we created boxplot distributions by month for TROPOMI $NO_2$ and HCHO composite values and NAM temperature composite values (**Figs. S5–S7**). Both TROPOMI HCHO VCDs and NAM 2-meter air temperatures follow a strong intra-seasonal cycle, but TROPOMI $NO_2$ VCDs do not. By comparing intra-seasonal differences in these monthly composites (**Figs. S5–S7**) to the differences we see between the $O_3$ season and CMA exceedance day composites (**Figs. 2f, 3c, and 5c**), we estimate that about 50 % of the HCHO and temperature changes are due to intra-seasonal changes (and the other 50 % due to $O_3$ exceedance day conditions) while 100 % of the $NO_2$ changes are due to exceedance day conditions (**Table S7**). More information about our methodology to separate intra-seasonal and episodic changes can be found in the text below **Table S7** in the supplemental information document.

In the supplemental information document, we added Figs. S5, S6, and S7, which show the intra-seasonal cycles of $NO_2$, HCHO and temperature, respectively. $NO_2$ does not appear to have a strong intra-seasonal cycle, while HCHO and 2-meter air temperatures do:

[Figure]

[Figure]

Our estimation methodology to separate intra-seasonal and episodic changes is shown in **Table S7**. The results show that 100 % of the $NO_2$ changes are due to exceedance day conditions while about 50 % of the HCHO and temperature changes are due to the intra-seasonal changes (and the other 50 % are due to exceedance day conditions):

|  | NO$_2$ [× 10$^{15}$ molec. cm$^{-2}$] | HCHO [× 10$^{15}$ molec. cm$^{-2}$] | Temperature [K] |
|---|---|---|---|
| (a) June–August mean | 1.747 (Fig. S3) | 8.96 (Fig. S4) | 297.83 (Fig. S5) |
| (b) May–September mean | 1.766 (Fig. S3) | 7.73 (Fig. S4) | 295.41 (Fig. S5) |
| (c) Difference [a − b] | -0.02 | 1.22 | 2.42 |
| (d) Mean difference between OS and Ex composites [Ex − OS] | 0.37 (Fig. 3c) | 2.52 molec. cm$^{-2}$ (Fig. 5c) | 4.31 (Fig. 2f) |
|  | NO$_2$ | HCHO | Temperature |
| (e) Percent of (d) due to monthly differences during the ozone season (intra-seasonal changes) approximated as: $[(e) = \frac{c}{d} \times 100\ \%]$ if (e) < 0 %, report as 0 % if (e) > 100 %, report as 100 % | 0 % | 48 % | 56 % |
| (f) Percent of (d) due to exceedance day differences (episodic events) approximated as: $[(f) = 100\ \% - (e)]$ | 100 % | 52 % | 44 % |

(2) If lake breeze circulation plays an important role in ozone pollution in the area, one may also expect differences in the vertical distribution of NO2 and HCHO between exceedance days and non-O3 days. This would lead to different biases in retrievals between the two composites and should be discussed.

**Response**

We agree that this is a valuable discussion to add to the manuscript.

**Changes to manuscript**

We added the following discussion to Section 3.1.1 on lines 272–277:

> One important thing to note is that because the strength of the lake breeze is different, it is possible that the vertical profiles of NO$_2$ and HCHO are different between exceedance days and non-exceedance days. The TROPOMI NO$_2$ and HCHO retrieval algorithms rely on forecasted model vertical profiles to produce VCD data (De Smedt et al., 2018; Van Geffen et al., 2022a). Therefore, the satellite retrievals used to create the ozone season and CMA exceedance day composites below (**Figs. 3-5**) may have different biases depending on how well the model forecasts vertical profiles of NO$_2$ and HCHO on exceedance days versus non-exceedance days in which the strength of the lake breeze circulation varies.

(3) Similarly, given the large differences in temperature (and the temperature-dependence of gas absorption cross sections), one would expect biases in retrievals on ozone exceedance days (if fixed cross sections are used in the slant column density fitting). This should be pointed out in the paper.

**Response**

We agree that this is valuable information to add to the manuscript.

According to the TROPOMI $NO_2$ ATBD (Van Geffen et al., 2022; https://sentinel.esa.int/documents/247904/2476257/Sentinel-5P-TROPOMI-ATBD-NO2-data-products), a correction factor is applied to the $NO_2$ absorption cross section to account for temperature sensitivity during the air mass factor step. The difference between the effective temperature of the $NO_2$ (in a specific layer) and the temperature of the baseline cross-section (220 K) is used to determine the correction factor assuming the temperature dependence is linear (equation 18 in the ATBD). Other gaseous species involved in the retrieval ($O_3$, $H_2O$, and the $O_2$-$O_2$ collision complex) use fixed cross sections at reference temperatures. However, it says in the ATBD that variations of these cross sections have little effect in the retrieval of $NO_2$ slant columns, which is why a correction factor is only applied for $NO_2$.

According to the TROPOMI HCHO ATBD (Hilboll et al., 2022; https://sentinels.copernicus.eu/documents/247904/2476257/Sentinel-5P-ATBD-HCHO-TROPOMI.pdf/db71e36a-8507-46b5-a7cc-9d67e7c53f70?t=1658313806426), the HCHO absorption cross section is used at a fixed reference temperature of 298 K. BrO, $NO_2$, and the $O_2$-$O_2$ collision complex also use cross sections at fixed temperatures. Only the $O_3$ cross section is adjusted by fitting two absorption cross sections at different temperatures and assuming a linear dependence on temperature.

**Changes to manuscript**

We added the following discussion to Section 3.1.1 on lines 290–302:

> Another important thing to note is that the significant differences in temperature between the ozone season and CMA exceedance day composites may also lead to different biases in the satellite retrievals used to create the $NO_2$ and HCHO composites below (**Figs. 3-5**). The absorption cross sections of various chemical species used in the TROPOMI retrieval algorithms are temperature dependent. This can lead to retrieval biases if the cross sections are not adjusted for temperature. To mitigate the potential bias for $NO_2$, a correction factor is applied to the $NO_2$ absorption cross section by calculating the difference between the effective temperature of the $NO_2$ and the temperature of the baseline cross section and assuming the temperature dependence is linear (Van Geffen et al., 2022). The other species used in the $NO_2$ retrieval algorithm ($O_3$, the $O_2$-$O_2$ collision complex, and $H_2O$) use fixed cross sections, but the temperature dependence of these cross sections has little effect in the retrieval of $NO_2$ (Van Geffen et al., 2022). In the HCHO retrieval algorithm, the cross sections of most of the species (HCHO, BrO, $NO_2$, and the $O_2$-$O_2$ collision complex) are fixed while the

O₃ cross section is adjusted by fitting two absorption cross sections at different temperatures and assuming a linear dependence on temperature (Hillboll et al., 2022). Retrieval biases stemming from using absorption cross sections at fixed temperatures may be larger in the CMA exceedance day composites of $NO_2$ and HCHO (**Figs. 3-5**) since temperatures tend to be warmer than usual as shown in **Figure 2**.

(4) Line 117: how does the change in spatial resolution of TROPOMI in 2019 (near the end of the ozone season) affect your results?

**Response**

We do not believe the change in TROPOMI spatial resolution affects our results in a substantial way because we composited data onto a 12 km × 12 km grid, which is a coarser spatial resolution than the TROPOMI pixel footprint (both before and after the upgrade to higher resolution). Therefore, we did not make any changes to the manuscript based on this question.

(5) Line 170: what are the noise levels of HCHO, NO2, and FNR at 4x4 km^2 and 12x12 km^2?

**Response**

We have provided calculations of relative "noise" levels between the 4 km and 12 km composites in the manuscript. The estimated "noise" level was 3.6 times higher for HCHO, 2.6 times higher for $NO_2$, and 3 times higher for FNR in the 4 km composites compared to the 12 km composites.

**Changes to manuscript**

We added the following text to the manuscript in Section 2.2 on lines 170–175:

Note that we used a grid with a coarser spatial resolution than the original TROPOMI pixel footprint based on a sensitivity test using a 4 km × 4 km grid. This sensitivity test involved identifying a region of fairly uniform HCHO and $NO_2$ in the 12 km composites and then comparing absolute differences between nearest neighbors ("noise") to the mean value within that region in both the 4 km and 12 km composites. The estimated "noise" level was 3.6 times higher for HCHO, 2.6 times higher for $NO_2$, and 3 times higher for FNR in the 4 km composites compared to the 12 km composites.

(6) Line 183: Are there differences in meteorology and chemical composition between more localized and more widespread ozone exceedance events?

**Response**

Yes, there may be differences. However, given the limited number of exceedance events, we do not feel there is enough data to produce statistically robust results/composites separately for more localized and more widespread ozone exceedance events. Therefore, we did not make any changes to the manuscript based on this question.

(7) Line 305: the coarse resolution of Kleipool climatology could be a bigger issue for urban areas – how does this affect your interpretation of the results over urban cores?

**Response**

This is a good question, and it is possible that the Kleipool climatology could be a bigger issue for urban areas. We do not know exactly how the coarse resolution affects retrievals over urban cores. However, the TROPOMI HCHO validation paper by De Smedt et al. (2021; https://doi.org/10.5194/acp-21-12561-2021) did compare TROPOMI HCHO to surface-based MAX-DOAS network column measurements, which includes instruments in major urban cores (e.g., Mexico City in Mexico, Madrid in Spain, Munich in Germany, and Beijing in China). The bias between TROPOMI and the MAX-DOAS measurements vary between sites, but it is found that on average TROPOMI HCHO observations are biased by -25 % for columns greater than $8 \times 10^{15}$ molec. cm$^{-2}$. We already noted this in Section 2.1 lines 140–141.

Our belief that there is an over water high HCHO bias for Lake Michigan is based on the fact that we see higher HCHO VCDs over all of the Great Lakes (Fig. S2). Our speculation that the Kleipool albedo climatology might be part of the cause of this artifact comes from a previous reviewer as well as the TROPOMI HCHO ATBD (Hilboll et al., 2022; https://sentinels.copernicus.eu/documents/247904/2476257/Sentinel-5P-ATBD-HCHO-TROPOMI.pdf/db71e36a-8507-46b5-a7cc-9d67e7c53f70?t=1658313806426), which states that the spatial resolution of the albedo dataset is coarser than the resolution of TROPOMI, which can induce errors in VCDs for coastal regions and is definitely something we should mention in the main text.

**Changes to manuscript**

We have expanded the discussion of the Kleipool climatology and its possible impact on HCHO observations over Lake Michigan in Section 3.1.2 on lines 332–342:

> Therefore, we believe this over water bias is unrealistic, especially since it is found over all the Great Lakes (**Fig. S2**). As noted in the TROPOMI HCHO algorithm theoretical basis document (Hilboll et al., 2022), the coarse resolution of the OMI-derived surface albedo climatology dataset used in the retrieval (Kleipool et al., 2008) can induce errors in VCD calculations for coastal regions. The high HCHO bias over

the Great Lakes may be in part due to the Kleipool dataset being too coarse to fully resolve the complex surface albedo properties that are common to lake surfaces. We also acknowledge that the resolution of the Kleipool climatology may also affect TROPOMI HCHO observations over urban cores since urban areas can also have complex surface albedo properties. However, TROPOMI validation studies (e.g., De Smedt et al., 2021) have been performed for urban sites, providing estimates of the TROPOMI retrieval biases for such areas (see **Section 2.1**). Validation over water surfaces is more difficult to conduct since the necessary instruments are not routinely deployed over lakes. Further research is needed to assess the impact of albedo changes between lake surfaces and the surrounding coastal areas on TROPOMI HCHO retrieval performance.

We additionally added **Figure S2** to the supplemental information document that shows the high HCHO artifact over the Great Lakes:

[Figure]

(8) Section 3.2: the weekday vs. weekend analysis appears to be only loosely connected to the previous section. I'm not sure if it actually adds any significant value to the paper.

**Response**

We consider the weekday versus weekend analysis to be a complement to the previous section because it shows how differences in $NO_2$ vertical column densities can also lead to substantial changes in FNR values (e.g., in the urban core of Chicago). This is opposite to the ozone season versus exceedance day analysis in which changes in FNR values are largely dominated by HCHO differences as opposed $NO_2$ differences.

**Changes to manuscript**

We have added the following text to the conclusions section on lines 569–575 to highlight the value that the weekday-weekend analysis adds to the paper:

> This weekday versus weekend analysis complements the ozone season versus exceedance day analysis because it shows that differences in $NO_2$ VCDs can also lead to substantial changes in FNR values and $O_3$ chemistry sensitivity (e.g., in the urban core of Chicago). This is opposite to the ozone season versus exceedance day analysis in which the largest changes in FNR values and $O_3$ chemistry sensitivity are dominated by HCHO differences. Additionally, opposite to the ozone season versus exceedance day analysis, we find no significant differences in 2-meter air temperature and 10-meter wind speed, direction, and divergence between weekdays and weekends.

(9) Section 4: it is not quite clear to me if there are any new insights gained from this study. Perhaps authors can emphasize any new results in the conclusions (and the abstract as well).

**Response**

We thank the referee for this comment, but we respectfully disagree due to the following:

- This work is important because the Lake Michigan region contains Chicago, which is one of the largest metropolitan areas in the United States that experiences ozone nonattainment associated with lake breeze circulations. The Lake Michigan region as a whole has been understudied compared to other regions (e.g., the Northeast U.S./New York City), especially regarding the use of satellite data. To our knowledge, our study is the first that utilizes TROPOMI data to assess changes in FNR values and inferred ozone chemistry sensitivities between ozone exceedance days and average ozone season days for the Lake Michigan region as a whole.
- One particularly new insight gained is that $NO_2$ concentrations appear to be concentrated in the urban core of Chicago on exceedance days due to the convergence of the wind field along the western Lake Michigan coastline. This result was discovered by connecting changes seen in the satellite based results to wind

divergence/convergence values calculated from model analysis meteorological data, which is a methodology not often performed in FNR studies. This study demonstrates the potential of using the model analysis wind data included in the TROPOMI data files to gain new insights into the transport patterns underlying the changes in chemical vertical column densities observed by TROPOMI.

- Another insightful finding is the possible high HCHO artifact over the Great Lakes, which deserves further investigation (already highlighted in the conclusions).
- Our results are comparable to another study conducted for New York City, which suggests that the results are applicable to other coastal urban environments with $O_3$ exceedance problems (already highlighted in the conclusions). Our study can be the basis of future work for other researchers who wish to conduct similar analyses for their areas of interest.

**Changes to manuscript**

To emphasize the value of this study and highlight some of the new insights gained, we added the following to the abstract:

Lines 14–16

Despite being a highly populated region with coastal $O_3$ air quality issues, the Lake Michigan region in the United States, including the Chicago, Illinois, metropolitan area (CMA), remains relatively understudied, especially from the satellite perspective. In this work, we present the first study that utilizes TROPOspheric Monitoring Instrument (TROPOMI) satellite data over the Lake Michigan region...

Lines 23–29

Utilizing 10-meter wind analysis data, we show that the lake breeze circulation is stronger on exceedance days. The strengthening of the lake breeze causes stronger convergence of the wind field along the southwestern Lake Michigan coastline, which can concentrate $NO_2$ emissions originating in this area. This finding provides a possible explanation for the higher TROPOMI $NO_2$ VCDs over the urban core of Chicago on exceedance days. Investigation of 2-meter air temperature analysis data reveals that temperatures are higher on exceedance days, which explains the stronger lake breeze circulation and provides a possible cause for the higher TROPOMI HCHO VCDs over the entire region (due to increased temperature dependent biogenic VOC emissions).

Similarly, we added the following to the conclusions:

Lines 539–543

Despite being a highly populated area that experiences coastal $O_3$ air quality problems, the Lake Michigan region is relatively understudied, especially from a satellite perspective. To address this research gap, we created mean formaldehyde to nitrogen dioxide ratio (HCHO/$NO_2$; "FNR") composites using 2019–2021 S5P TROPOMI

satellite data over the Lake Michigan region to assess changes in ozone precursor levels and the inferred $O_3$ chemistry sensitivity between: (1) $O_3$ season days and Chicago metropolitan area (CMA) $O_3$ exceedance days, and (2) weekdays and weekends.

Lines 552–560

Ten-meter wind analysis data shows that the lake breeze circulation along the southwestern Lake Michigan coastline is stronger during CMA exceedance days, which causes stronger convergence of the wind field and the concentration of $NO_2$ emissions originating in the area. Thus, the strengthening of the lake breeze is a possible cause for the higher TROPOMI composite $NO_2$ VCDs in the urban core of Chicago on exceedance days. This analysis demonstrates the potential of using the model analysis wind data included in the TROPOMI data files to gain new insights into the transport patterns underlying the changes in chemical vertical column densities observed by TROPOMI. Both higher TROPOMI HCHO composite VCDs and the stronger lake breeze can be explained by higher temperatures on exceedance days, which we showed to be true using model analysis 2-meter air temperature data.

**Response to Referee #4**

**Referee's introductory comment**

The revised manuscript is much improved over the original version. Major areas of improvement include the following:
- Use of PAMS data has been eliminated from the manuscript. The 6-day integrated HCHO measurements are of little or no value for use in HCHO/NO2 ratio analysis.
- Explanation of the uncertainty in use of the J20 criteria values for NOx sensitive and VOC sensitive regimes has been included. Use of these criteria are especially justified in examining high O3 exceedance days.
- The same number of weekday and weekend days are now used in the analysis.
- It has been demonstrated that useful results are obtained even though the analysis period covers the Covid-19 pandemic lockdown. The same general results for ozone sensitivity are found for each of the three years.
- Section 2.4 has been added, which covers the J20 criteria uncertainty, the differences in OMI vs. TROPOMI, and the Covid-19 effect.

It is good to see that reprocessed TROPOMI NO2 data have been used in the revised version of the manuscript. Now all NO2 data used in the calculations are from the same version of the retrieval algorithm.

The paper should be published after minor revision.

**Response to introductory comment**

We thank referee #4 for reviewing our revised manuscript. Based on both referees' comments, we have made the following new revisions to the manuscript:

1. Adding an analysis to estimate how much of the differences between the ozone season and exceedance day composites are due to intra-seasonal variations versus episodic changes on exceedance days,
2. Expanding discussions of: (a) possible satellite retrieval biases due to vertical profile and temperature differences on exceedance days, (b) the possible impact of the coarse albedo climatology dataset on the TROPOMI high HCHO artifact over the Great Lakes, and (c) "noise" levels of 4 km and 12 km versions of the TROPOMI composites,
3. Including a comparison of AQS surface $NO_2$ measurements to our TROPOMI-based results, and
4. Emphasizing new insights gained from and the value of this study in the abstract and the conclusions sections.

Our response to referee #4's specific comment is as follows:

**Referee's specific comment**

(1) Concerning the larger NO2 seen over the whole domain on ozone exceedance days: I would recommend that the authors determine if surface NO2 monitoring network data show the same result. This analysis could very easily be done.

**Response**

We agree that looking at surface $NO_2$ monitoring network data would be a great comparison to the TROPOMI-based results.

We analyzed U.S. EPA Air Quality System (AQS) surface $NO_2$ data at 13:00 local time (approximately matching the TROPOMI overpass time) for 8 monitoring sites within the study domain. These 8 sites were selected as they were the only ones that had data for our entire 3-year study period from 2019–2021.

**Changes to manuscript**

We added **Table S5** in the supplemental information document that compares the ozone season mean values to the exceedance day mean values. For 7 of the 8 sites, surface $NO_2$ levels are higher on exceedance days compared to the ozone season. When the 8 sites are averaged together, we find that surface $NO_2$ levels on exceedance days are higher by about 19 % compared to the ozone season average. This closely matches the TROPOMI composite results in **Figure 3c**, which shows a domain-wide average $NO_2$ vertical column density increase of about 21 % on exceedance days:

| Site* | Ozone season mean (OS) [ppb] | Exceedance days mean (Ex) [ppb] | Ex – OS [ppb] | Ex % change from OS |
|---|---|---|---|---|
| A (17_31_119) | 13.01 | 17.69 | +4.68 | +36 % |
| B (17_31_219) | 8.94 | 10.21 | +1.27 | +14 % |
| C (17_31_3103) | 10.77 | 13.16 | +2.39 | +22 % |
| D (17_31_4002) | 6.65 | 7.20 | +0.55 | +8 % |
| E (17_31_76) | 5.65 | 5.98 | +0.33 | +6 % |
| F (18_141_15) | 2.17 | 1.45 | -0.72 | -33 % |
| G (18_89_22) | 2.71 | 2.76 | 0.05 | + 2 % |
| H (55_79_56) | 7.69 | 9.04 | 1.35 | +18 % |
| **All sites** | **7.27** | **8.62** | **1.35** | **+19 %** |
| **TROPOMI composite – full domain** | **1.75 molec. cm$^{-2}$** | **2.11 molec. cm$^{-2}$** | **0.36 molec. cm$^{-2}$** | **+21 %** |

AQS surface $NO_2$: ozone season vs. CMA exceedance days

We additionally included a map in the supplemental information document (**Fig. S1**) that shows the locations of the 8 surface monitors:

[Figure]

Finally, in the main text we added the following sentence to Section 3.1.2 on lines 315–317:

We note that surface $NO_2$ observations at 13:00 local time from the average of eight AQS monitoring sites also indicate higher $NO_2$ levels on CMA exceedance days compared to the average across all $O_3$ season days during the 2019–2021 study period (**Table S5; Fig. S1**).